behaviour

interspecies comparison, isolation-induced call, rodent, ultrasonic vocalization, morphometric variables, body weight

**Author for correspondence:**
Ilya A. Volodin
e-mail: volodinsvoc@gmail.com

# Pup ultrasonic isolation calls of six gerbil species and the relationship between acoustic traits and body size

Julia D. Kozhevnikova[1], Ilya A. Volodin[1,2], Alexandra S. Zaytseva[1,3], Olga G. Ilchenko[3] and Elena V. Volodina[2]

[1]Department of Vertebrate Zoology, Faculty of Biology, Lomonosov Moscow State University, Vorobievy Gory, 1/12, 119234 Moscow, Russia
[2]Department of Behaviour and Behavioural Ecology, A.N. Severtsov Institute of Ecology and Evolution, Russian Academy of Sciences, Moscow 119071, Russia
[3]Department of Small Mammals, Moscow Zoo, Moscow, Russia

IAV, 0000-0001-6278-0354

Among Gerbillinae rodents, ultrasonic calls of adults of small-sized species are typically higher frequency than those of adults of large-sized species. This study investigates whether a similar relationship can be found in pups of six gerbil species (*Dipodillus campestris*, *Gerbillus perpallidus*, *Meriones unguiculatus*, *Meriones vinogradovi*, *Sekeetamys calurus* and *Pachyuromys duprasi*). We compared the average values of acoustic variables (duration, fundamental and peak frequency) of ultrasonic calls (20 calls per pup, 1200 in total) recorded from 6- to 10-day-old pups (10 pups per species, 60 in total) isolated for 2 min at 22°C and then weighed and measured for body variables. The longest calls (56 ± 33 ms) were found in the largest species, and the highest frequency calls (74.8 ± 5.59 kHz) were found in the smallest species. However, across species, call duration (ranging from 56 to 159 ms among species) did not display a significant relationship with pup body size; and, among frequency variables, only the minimum fundamental frequency depended on pup body size. Discriminant analysis assigned 100% of calls to the correct species. The effect of species identity on the acoustics was stronger than the effect of body size. We discuss these results with the hypotheses of acoustic adaptation, social complexity, hearing ranges and phylogeny.

## 1. Introduction

Gerbils or jirds (Rodentia, Gerbillinae) are a subfamily comprising approximately 15 genera with 81 species of small desert

mammals [1,2]. Gerbillinae species represent a traditional model for comparative studies of social behaviour [3–12], locomotion [3,13,14], ecological adaptations [3,15,16], physical development [3,17–21], interspecies hybridization [22–24] and call-based taxonomy [7,25–27]. Many species of gerbils have been kept in captivity [3,20,21,28], and some of them serve as wild-type (i.e. not genetically modified or selected for behaviour) animal models for biomedical research [28–44].

Adult gerbils, depending on species, are known to vocalize in the ultrasonic range of frequencies, e.g. most South African gerbils [27]; or in the audible range of frequencies, e.g. Middle Asian great gerbils *Rhombomys opimus* [45–48], Egyptian pale gerbils *Gerbillus perpallidus* [46] and Namibian dune hairy-footed gerbils *Gerbillurus tytonis* [3]. Some species vocalize in both audible and ultrasonic ranges, e.g. Mongolian gerbils *Meriones unguiculatus* [49,50], Egyptian fat-tailed gerbils *Pachyuromys duprasi* [25,51,52] and Tunisian fat sand rats *Psammomys obesus* [25].

Infant gerbils vocalize primarily in the ultrasonic range of frequencies, e.g. Mongolian gerbils *Meriones unguiculatus* [53–56], fat-tailed gerbils *Pachyuromys duprasi* [51,52,57] and gerbils of the genus *Gerbillurus*: *Gerbillurus vallinus* and *Gerbillurus setzeri* [18]. Infant gerbil ultrasonic vocalizations (USVs) trigger parental pup retrieval behaviour, investigated in the Mongolian gerbil [58–61] and in two *Gerbillurus* species [3]. In addition to communication with parents, pup USVs can function for improving thermoregulation via better oxygenation of brown fat during the calling [62]. Another hypothesis suggests that USVs can be by-products of cold-affected abdominal compression, affecting blood circulation [63,64]. In Muridae rodents, pup isolation USVs are triggered by temperatures below 33°C (approximately nest temperature) [65]; vocal activity increases in response to both hypothermia and hyperthermia [66].

Gerbil species are different in body size [3,15,20]. Among Gerbillinae species, the body mass of adults can differ more than 15 times: approximately 10 g in *Gerbillus henleyi* [15,67], approximately 19 g in *Gerbillus simoni* [68], approximately 24 g in *Gerbillus dasyurus* [15], approximately 90 g in *Meriones crassus* [15,67] and over 150 g in *Rhombomys opimus* and *Psammomys obesus* [67,69]. For many gerbil species, morphometric data (body weight, body length, head length and foot length) were used as proxies of body size in both pups and adults [3,17–21,52].

A substantial variation in body size enables to directly test this hypothesis about the effects of body size on the acoustics of ultrasonic calls across Gerbillinae species. In adult gerbils, the USV fundamental frequency (f0) overall displays a negative correlation with species' body size among six South African species of gerbils: the small-sized *Gerbillurus paeba* and *Gerbillurus tytonis* produce higher frequency USVs compared with the large-sized *Gerbillurus setzeri* and *Gerbillurus vallinus* [27]. However, two other species, *Gerbilliscus leucogaster* and *Desmodillus auricularis*, both vocalize at frequencies similar to the smaller *Gerbillurus setzeri* and *Gerbillurus vallinus* [27].

For pup gerbils, USVs have yet to be examined in cross-species perspective and for their potential relationship between f0 and body size. In this study, USVs of pups of six gerbil species from Asia and North Africa (*Dipodillus campestris*, *Gerbillus perpallidus*, *Meriones unguiculatus*, *Meriones vinogradovi*, *Sekeetamys calurus* and *Pachyuromys duprasi*) were investigated for their acoustic variables. These species differ in distribution areas and body size. The North African gerbil (*Dipodillus campestris* Levaillant, 1857), with body mass from 32 to 38 g in the wild [70], has a broad distribution area in North Africa [71]. The pale gerbil (*Gerbillus perpallidus* Setzer, 1958), with body mass of approximately 30 g in the wild [72], has a narrow distribution area to the east of the Nile river main bed and the Nile river delta in North Africa [73]. The Mongolian gerbil (*Meriones unguiculatus* Milne-Edwards, 1867), with body mass of approximately 45–65 g in the wild [74], lives in far east of Russia, in Mongolia and in northeast China [74,75]. The Vinogradov's gerbil (*Meriones vinogradovi* Heptner, 1931), with body mass in captivity of approximately 154 g (I.A.V. 2018, unpublished data), is distributed in Turkey, Syria, Iran, Iraq, Armenia and Azerbaijan [76,77]. The bushy-tailed gerbil (*Sekeetamys calurus* Ellerman, 1947), with body mass from 34 to 58 g in the wild [67,78,79], is distributed in the Middle East and eastern Egypt [80,81]. The fat-tailed gerbil (*Pachyuromys duprasi* Lataste, 1880), with body mass of approximately 36.5 g in the wild [80] and approximately 60–80 g in captivity [21,28], is distributed in North Africa.

Among the six study species, the USVs are reported in pup *Meriones unguiculatus* [53–56], adult *Meriones unguiculatus* [49,50], pup *Pachyuromys duprasi* [21,51,52] and in adult *Pachyuromys duprasi* [25,51,52]. In *Gerbillus perpallidus*, only audible calls of adults were earlier investigated [46]. The aim of this study was to examine the isolation-induced USVs of 6–10-day-old pups of six gerbil species for their species-specific and shared features and to reveal the relationship between the USV acoustic traits and body size.

# 2. Material and methods

## 2.1. Study site and subjects

The isolation-induced pup USVs were collected in six captive colonies of gerbils at Moscow Zoo, Moscow, Russia. The *Dipodillus campestris* colony originated in 1989 from four individuals from the Museum of Natural History, France; the *Gerbillus perpallidus* colony originated in 1985 from three individuals obtained from German zoos; the *Meriones unguiculatus* colony originated in 2009 from 11 individuals from a natural colony in Tuva, Russia; the *Meriones vinogradovi* colony originated in 2006 from a natural colony (Armenia); the *Sekeetamys calurus* colony originated in 2002 from four individuals from zoos (Germany) and four individuals obtained in 2009 from a natural colony in the United Arab Emirates; and the *Pachyuromys duprasi* colony originated in 2007 from eight individuals from Egypt.

The USVs of five of the six species (*Dipodillus campestris*, *Gerbillus perpallidus*, *Meriones unguiculatus*, *Meriones vinogradovi* and *Sekeetamys calurus*) were collected in February–July 2018, and the calls of the sixth species (*Pachyuromys duprasi*) were collected in May–July 2013–2014. While pup *Meriones unguiculatus* produce USV calls from the first day of life [53,54,56], pup *Pachyuromys duprasi* produce USVs only from the fifth day of life [51,52,57]. So, we selected the pup age class of 6–10 days old for cross-species comparative analyses.

Before parturition, females of the captive colonies were checked three times per week for the appearance of a litter, and birth dates, as well as the number of pups, were recorded. The day of birth was considered day zero of a pup's life. The pups originated from 36 litters (5–8 litters per species) delivered by 36 breeding pairs of the six species. We took 1–3 pups per litter. Pups were unsexed; in each litter pups were selected randomly. In total, we included in the analysis USVs of the 60 pups, 10 pups per species.

## 2.2. Animal housing

The animals were kept under a natural light regime at room temperature (24–26°C), in family groups consisting of two parents and littermates, because males of these species are non-aggressive to pups [20,21,51]. The animals were housed in wire-and-glass cages of $51 \times 42.5 \times 41.5$ cm or $40 \times 100 \times 40$ cm depending on animal size and group size, with bedding of sawdust and hay, various shelters, cardboard boxes and tree branches as enrichment. They received custom-made small desert rodent chow with insect and mineral supplements and fruits and vegetables ad libitum as a source of water.

## 2.3. Experimental procedure and ultrasonic vocalization recording

All acoustic recordings were conducted in a separate room where no other animals were present, at a room temperature of 22–25°C during the daytime, at the same level of background noise for all tested animals, and electric lamps and powered equipment switched off. For USV recordings (sampling rate 384 kHz, 16-bit resolution), we used a Pettersson D1000X recorder with built-in microphone (Pettersson Electronik AB, Uppsala, Sweden). The microphone was placed stationary at a distance of 35 cm above the tested animal. The obtained recordings had a high signal/noise ratio, with calls not masked with background noise and minimal reverberation. For visualizing the USVs during the experimental trials, we also used the Echo Meter Touch 2 PRO (Wildlife Acoustics, Inc., Maynard, MA, USA), which allowed tracking of USV spectrogram in real time on a compatible smartphone display.

Each subject animal participated only in one experimental trial. Each individual was tested singly. The audio track of a focal animal during the experimental trial was recorded as a wav-file. Immediately before an experimental trial, the focal animal was taken from the home cage and transferred in a small clean plastic hutch to the experimental room within the same floor of the building. Time from the removal of the focal animal from the cage to the start of an experimental trial did not exceed 60 s. During the trial, the animal was isolated on a clean plastic tray $190 \times 130 \times 70$ mm, standing on an even plastic table surface. The recording started when the focal animal was placed in the experimental set-up and lasted 120 s. No additional interventions were made by the experimenter and animals could move freely. We did not measure pup body temperature, to minimize the manipulations on the pup during the trial.

After the trial, the focal animal was weighed and measured for body length, head length, foot length and tail length. For weighing, we used G&G TS-100 electronic scales (G&G GmbH, Neuss, Germany),

accurate to 0.01 g. The weighing was done in the same plastic hutch which was used for transferring the animal to the experimental set-up. For the length measurements, we used electronic callipers (Kraf Tool Co., Lenexa, KS, USA), accurate to 0.01 mm. We measured body length of the hand-held animal from the tip of the snout to the anus, and head length from the tip of the snout to the occiput. We measured foot length from the heel to the tip of the middle toe, and tail length from the anus to the tip of the tail. These measurements were repeated three times and the mean value was taken for analysis. The body variables and log body weight were taken as proxies of body size for further comparison with the USV acoustic variables.

If more than one littermate per litter was tested, after the end of a trial, the focal pup was moved to a heated hutch on a bedding of cotton fabric in a neighbouring room. Experimental trials with all focal littermates were done one by one in the same manner. Then, all of them were simultaneously returned to their home cage to their parents; the time pups spent away from their nest did not exceed 30 min. The experimental set-up was rubbed with a napkin wetted with alcohol after each experimental trial, to avoid the effect of smell on USVs of subsequent focal animals [82–84].

## 2.4. Call samples

We use the term 'isolation call' to refer to any ultrasonic call produced by an individual pup during the 2-min experimental isolation procedure. Using visual inspection of spectrograms of acoustic files created with Avisoft SASLab Pro software (Avisoft Bioacoustics, Berlin, Germany), we selected 20 USVs per individual, taking calls randomly from those with high signal-to-noise ratio and without superimposed noise from different parts of each 120 s recording, approximately one ultrasonic call per 5–6 s, avoiding taking calls following each other. Call frequency contour shape and the presence of nonlinear phenomena were not considered as selection criteria. Following [51], we defined ultrasonic call as frequency contour either continuous without breaks or with breaks shorter than 10 ms and frequency jumps less than 10 kHz. If the separation break exceeded 10 ms, we considered that the contours belonged to two different calls. In total, we included in analyses 200 USVs per species, 1200 USVs in total.

## 2.5. Acoustic analysis

Measurements of acoustic variables of pup USVs were conducted with Avisoft and automatically exported to Microsoft Excel (Microsoft Corp., Redmond, WA, USA). As (based on the visual inspection of spectrograms) the minimum f0 of ultrasonic calls always exceeded 10 kHz, all wav-files were subjected to 10 kHz high-pass filtering before measurements to remove low-frequency noise. For each USV, we measured, in the spectrogram window of Avisoft (sampling frequency 384 kHz, Hamming window, FFT 1024 points, frame 50%, overlap 87.5%, providing frequency resolution 375 Hz and time resolution 0.33 ms), the duration with the standard marker cursor, the maximum f0 (f0max), the minimum f0 (f0min), the f0 at the onset of a call (f0beg) and the f0 at the end of a call (f0end) with the reticule cursor (figure 1 and electronic supplementary material, table S1). For each USV, we measured, in the power spectrum window of Avisoft, the frequency of maximum amplitude (fpeak) from the call's mean power spectrum (figure 1 and electronic supplementary material, table S1).

## 2.6. Ultrasonic vocalization contour shapes and nonlinear vocal phenomena

In the spectrogram window of Avisoft, we classified the ultrasonic calls manually according to the five f0 contour shapes: flat, chevron, upward, downward and complex (figure 2 and electronic supplementary material, audio S1). One researcher (J.D.K.) classified the calls and another researcher (I.A.V.) confirmed this classification. This classification was based (with modifications) on classifications developed for domestic mice *Mus musculus* [85–87], fat-tailed gerbils [51] and yellow steppe lemmings *Eolagurus luteus* [88]. The flat contour was denoted when the difference between f0 min and f0max was less than 6 kHz. When the difference between f0 min and f0max exceeded 6 kHz, the denoted contours could be the chevron (up-down one time), upward (ascending from start to end), downward (descending from start to end) or complex (up-down many times or U-shaped).

For each USV, we noted the presence of nonlinear vocal phenomena (figure 3 and electronic supplementary material, audio S2): frequency jumps and biphonations [51,88–91]. Frequency jump was denoted when f0 suddenly changed for ≥10 kHz up or down (figure 3) and following [51,85–88]. In the previous study on the fat-tailed gerbils, the USVs with one frequency jump were termed

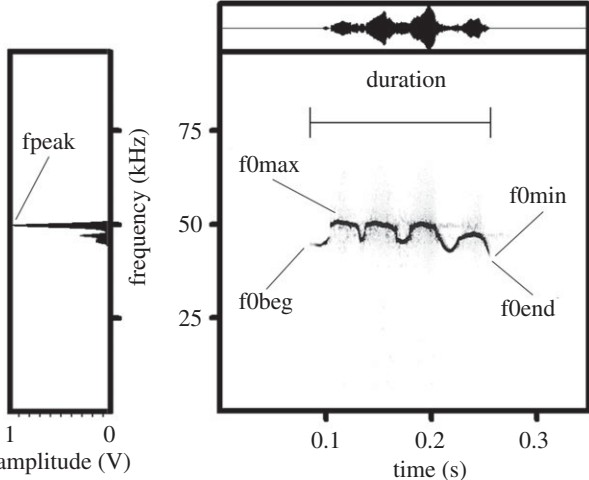

**Figure 1.** Measured variables for pup gerbils ultrasonic calls exemplified by a call with complex contour of *Meriones vinogradovi*. Spectrogram (right) and mean power spectrum of the entire call (left). Designations: duration, call duration; f0beg, the fundamental frequency at the onset of a call; f0end, the fundamental frequency at the end of a call; f0max, the maximum fundamental frequency; f0 min, the minimum fundamental frequency; fpeak, the frequency of maximum amplitude. The spectrogram was created using a sampling frequency of 192 kHz, Hamming window, FFT 1024 points, frame 50% and overlap 93.75%.

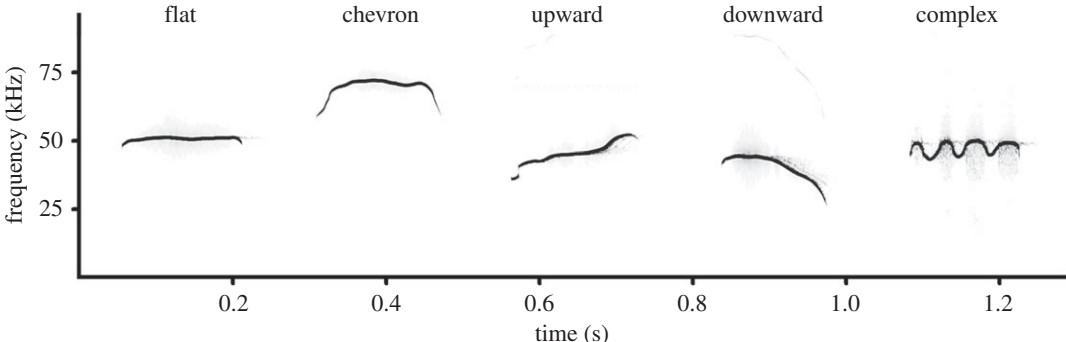

**Figure 2.** Spectrogram illustrating the five contour shapes occurring in pup ultrasonic calls. From left to right: pup USV call with flat contour of *Gerbillus perpallidus*; with chevron contour of *Dipodillus campestris*; with upward contour of *Meriones unguiculatus*; with downward contour of *Sekeetamys calurus* and with complex contour of *Meriones vinogradovi*. The audio file is available in the electronic supplementary material, audio S1. The spectrogram was created using a sampling frequency of 192 kHz, Hamming window, FFT 1024 points, frame 50% and overlap 87.5%.

'two-note' calls and the USVs with more number of frequency jumps were termed 'multi-note' calls [51]. Biphonation was denoted when two independent fundamental frequencies, the low (f0) and the high (g0) and their combinatory frequency bands (g0–f0, g0–2f0, etc.) were found in an ultrasonic call [85,88] (figure 3). Following [88], we determined the type of the frequency contour with frequency jumps by virtually smoothing the contour as if the fundamental frequency trace was uninterrupted throughout a call.

## 2.7. Statistical analyses

Statistical analyses were conducted using STATISTICA, v. 8.0 (StatSoft, Tulsa, OK, USA) and R v. 3.0.1 [92]. All means are given as mean ± s.d. Significance levels were set at 0.05, and two-tailed probability values are reported. For each individual subject, the averaged values of six acoustic variables over 20 calls were used for the statistical comparisons. This allowed avoiding multiple measurements of acoustic variables from the same individual (pseudoreplication), to match one averaged acoustic measurement per individual with one measurement of each body variable per individual and to

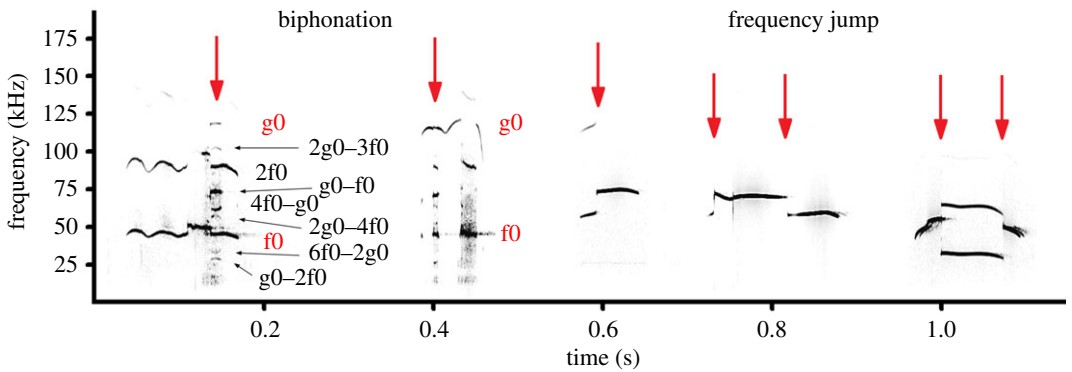

**Figure 3.** Spectrogram illustrating the nonlinear phenomena occurring in pup ultrasonic calls. From left to right: two pup USV calls with biphonation (*Meriones vinogradovi*); three pup USV calls with frequency jumps: with frequency jump up (*Dipodillus campestris*); with frequency jump up-down (*Dipodillus campestris*) and with frequency jump down-up (*Pachyuromys duprasi*). Designations: f0, the low fundamental frequency band (45 kHz in the first call); g0, the high fundamental frequency band (118.5 kHz in the first call); 2f0, harmonic of f0, g0–f0, g0–2f0, 2g0–3f0, 2g0–4f0, 4f0–g0 and 6f0–2g0—combinatory frequency bands. Red arrows indicate call sections with biphonation or points of frequency jumps. The audio file is available in the electronic supplementary material, audio S2. Spectrogram was created using a sampling frequency of 384 kHz, Hamming window, FFT 1024 points, frame 50% and overlap 75%.

decrease the number of degrees of freedom for more robust results. The values were normally distributed for all body size and acoustic variables (Kolmogorov–Smirnov test), so we could use the parametric tests.

We used one-way ANOVA with Tukey HSD (Tukey's honest significant difference) test to estimate the effects of species identity on the variables of body size. We used principal component analysis (PCA) to estimate the degrees of correlation between body size variables and for calculating the body size index on the basis of these variables. We used generalized linear mixed model (GLMM) with Tukey HSD test for estimating the effects of pup species and pup body size on the acoustic variables of USVs, with pup species as a categorical factor and index body size as a continuous factor. We used binomial GLM to compare the presence or absence of nonlinear vocal phenomena between species, with pup species as a categorical factor and the presence or absence of nonlinear vocal phenomena as a response factor.

We used Pearson correlation with body size index as a proxy of body size to estimate the effect of body size on the acoustics of USV calls. We used the discriminant function analysis (DFA) standard procedure based on the six measured acoustic variables to estimate the potential for distinguishing species by USVs of gerbil pups. We performed a cross-validated (leave-one-out) DFA to determine if USVs could be correctly classified to the correct species. Variables contributing most to discrimination were established using Wilks' lambda [93,94].

# 3. Results

## 3.1. Body size

The precise age (in days) of subject 6–10-day-old pups on the dates of their experimental trials did not differ between species, excluding *Pachyuromys duprasi*, which were significantly older than *Meriones vinogradovi* or *Seeketamus calurus* pups (table 1). Pups of *Meriones vinogradovi* were significantly heavier than pups of any other species and had longer heads than in other species, except for *Sekeetamus calurus*. Body mass differed significantly between *Dipodillus campestris* and *Meriones vinogradovi* and between *Dipodillus campestris* and *Gerbillus perpallidus*. Body length was shorter in *Dipodillus campestris* than in *Pachyuromys duprasi* or *Meriones vinogradovi* and did not differ otherwise between species. Foot length was similar in all species. Tail length was the shortest in *Pachyuromys duprasi* (table 1).

For calculating the body size index, we took body mass, body length and head length. We excluded foot length as it varied only slightly between species and tail length as its variation did not reflect body size of study species (table 1). Body mass, body length and head length were correlated with the first

**Table 1.** Values (mean ± s.d.) for pup precise age (days) and body size variables of 6–10-day-old pups of six Gerbillinae species and one-way ANOVA results for their comparison between species. Designations: $N$, the number of individual pups; D.c., *Dipodillus campestris*; G.p., *Gerbillus perpallidus*; M.u., *Meriones unguiculatus*; M.v., *Meriones vinogradovi*; S.c., *Sekeetamys calurus*; P.d., *Pachyuromys duprasi*. The same letters next to values indicate the lack of significant differences between them ($p > 0.05$, Tukey *post hoc*).

| variable | D.c., N = 10 | G.p., N = 10 | M.u., N = 10 | M.v., N = 10 | S.c., N = 10 | P.d., N = 10 | ANOVA |
|---|---|---|---|---|---|---|---|
| pup precise age (days) | 7.8 ± 0.9$^{a,b}$ | 7.4 ± 0.7$^{a,b}$ | 6.9 ± 1.4$^{a,b}$ | 6.8 ± 0.4$^{a}$ | 6.6 ± 0.5$^{a}$ | 8.1 ± 1.3$^{b}$ | $F_{5,54} = 4.05; p = 0.003$ |
| body mass (g) | 5.43 ± 1.67$^{a}$ | 7.34 ± 2.39$^{b}$ | 6.79 ± 0.75$^{a,b}$ | 9.43 ± 1.52$^{c}$ | 7.08 ± 0.86$^{a,b}$ | 6.60 ± 0.58$^{a,b}$ | $F_{5,54} = 8.32; p < 0.001$ |
| body length (mm) | 42.8 ± 6.3$^{a}$ | 47.1 ± 5.6$^{a,b}$ | 44.2 ± 2.4$^{a,b}$ | 49.7 ± 4.9$^{b}$ | 45.2 ± 2.4$^{a,b}$ | 49.3 ± 1.6$^{b}$ | $F_{5,54} = 4.31; p = 0.002$ |
| head length (mm) | 18.1 ± 2.2$^{a,d}$ | 20.4 ± 2.1$^{b,d}$ | 19.4 ± 1.2$^{a,d}$ | 22.8 ± 1.3$^{c}$ | 21.6 ± 1.0$^{b,c}$ | 20.1 ± 0.9$^{a,b,d}$ | $F_{5,54} = 11.44; p < 0.001$ |
| foot length (mm) | 14.3 ± 3.2$^{a,b}$ | 16.4 ± 3.3$^{b}$ | 13.2 ± 1.2$^{a}$ | 14.8 ± 1.4$^{a,b}$ | 13.4 ± 1.1$^{a}$ | 12.7 ± 0.9$^{a}$ | $F_{5,54} = 3.87; p = 0.005$ |
| tail length (mm) | 26.5 ± 9.1$^{a,b}$ | 28.7 ± 8.7$^{a,b}$ | 21.8 ± 2.3$^{a}$ | 26.7 ± 2.8$^{a,b}$ | 30.5 ± 3.8$^{b}$ | 9.7 ± 0.7$^{c}$ | $F_{5,54} = 18.49; p < 0.001$ |

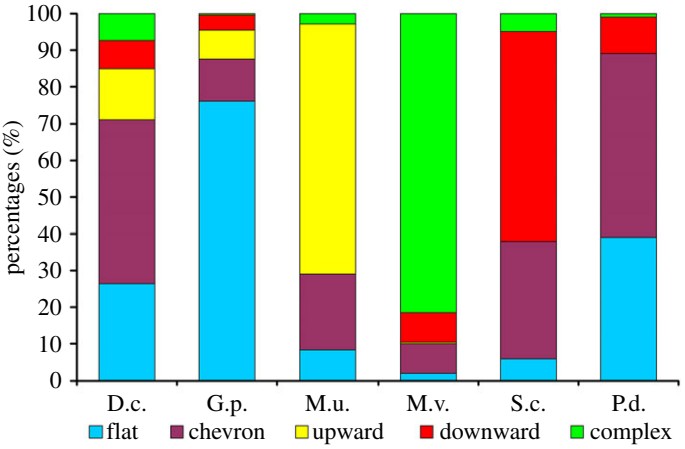

**Figure 4.** Percentages of five different USV contour shapes in the total sample of 1200 ultrasonic calls from 60 individual 6–10-day-old pups of six Gerbillinae species (10 pups per species, 20 calls per pup). Designations: D.c., *Dipodillus campestris*; G.p., *Gerbillus perpallidus*; M.u., *Meriones unguiculatus*; M.v., *Meriones vinogradovi*; S.c., *Sekeetamys calurus*; P.d., *Pachyuromys duprasi*. Contour names are provided in the figure.

**Table 2.** Correlation coefficients between body size variables and PCA factors, eigenvalues and per cent variance, described by each PCA factors.

| variable | PCA factor 1 | PCA factor 2 | PCA factor 3 |
|---|---|---|---|
| body mass | −0.952 | 0.140 | 0.273 |
| body length | −0.913 | −0.405 | −0.049 |
| head length | −0.940 | 0.252 | −0.229 |
| eigenvalue | 2.623 | 0.247 | 0.129 |
| per cent variance (%) | 87.45 | 8.24 | 4.31 |

PCA factor very highly, with correlation coefficients from 0.91 to 0.95 (table 2). The first PCA factor accounted for 87.45% of the variation. Then, we used the values of the first PCA factor for each of 120 pups as a generalizing body size index.

## 3.2. Call categories

In the total sample of 1200 USVs taken from the total of 60 subjects of the six species, the most abundant call contour shape was chevron: 333 USVs (27.8%), followed by flat contour: 316 USVs (26.3%), complex: 197 USVs (16.4%), upward: 181 USVs (15.1%) and downward: 173 USVs (14.42%). Species differed substantially in their most frequently used contour shape of USVs. *Gerbillus perpallidus* primarily produced USVs with flat contour (76% of 200 calls), *Meriones unguiculatus* with upward contour (68%), *Meriones vinogradovi* with complex contour (81.1%) and *Sekeetamus calurus* with downward contour (57%). Both *Pachyuromys duprasi* and *Dipodillus campestris* primarily produced USVs with chevron contour (50% and 44.5%, respectively) (figure 4).

Nonlinear phenomena were absent in pup USVs of *Gerbillus perpallidus* but were common in *Pachyuromys duprasi* and rare in the other four gerbil species (figure 5). Binomial GLM showed that species identity significantly affected the presence or absence of nonlinear vocal phenomena in pup USVs (estimate = 0.472 ± 0.076, $z = 6.24$, $p < 0.001$). Within the total sample of 1200 USVs of all 60 subjects of six species, the most common nonlinear phenomenon was frequency jump, which was present in 84 (7.0%) USVs; biphonations were rare, presented only in seven (0.6%) USVs. No one single ultrasonic call contained both frequency jump and biphonation. Frequency jumps were present in 29% USVs of *Pachyuromys duprasi*, 9.5% USVs of *Dipodillus campestris*, 3% USVs of *Meriones vinogradovi* and 0.5% USVs (one of 200) of *Sekeetamys calurus*. Biphonations were present in 2.5% USVs of *Meriones vinogradovi* and in 1% USVs of *Meriones unguiculatus* (figure 5).

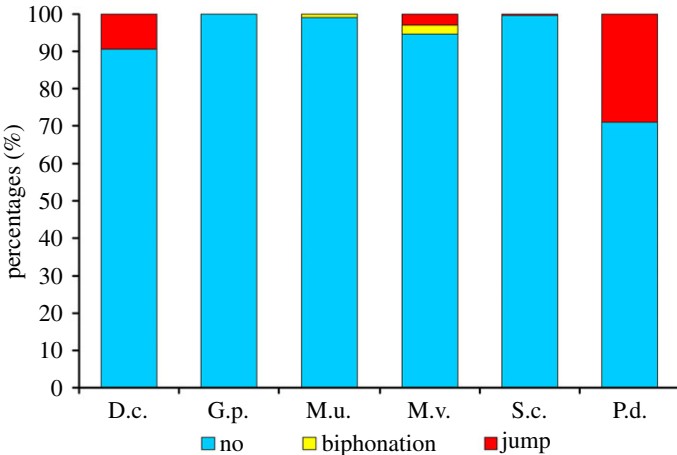

**Figure 5.** Percentages of two different nonlinear phenomena in the total sample of 1200 ultrasonic calls from the 60 individual 6–10-day-old pups of six Gerbillinae species (10 pups per species, 20 calls per pup). Designations: D.c., *Dipodillus campestris*; G.p., *Gerbillus perpallidus*; M.u., *Meriones unguiculatus*; M.v., *Meriones vinogradovi*; S.c., *Sekeetamys calurus*; P.d., *Pachyuromys duprasi*.

## 3.3. Acoustic variables

The duration of pup ultrasonic calls was the longest in *Meriones vinogradovi*, which differed significantly from those in all species except for *Sekeetamys calurus* (table 3 and figure 6). The shortest duration was found in *Pachyuromys duprasi*, which differed significantly from the duration of all species except *Gerbillus perpallidus*. USVs of the remaining species were of intermediate duration and did not differ significantly by this variable (table 3 and figure 6).

The highest f0max was found in USVs of pup *Dipodillus campestris*; it was significantly different from the f0max of all other species. The lowest f0max was found in USVs of pup *Sekeetamys calurus*; it was significantly different from those of all other species except for *Meriones unguiculatus* (table 3 and figure 6).

The highest f0 min was also found in pup USVs of *Dipodillus campestris*; it was significantly different from the f0 min of all other species. The slightly lower f0 min was found in pup USVs of *Gerbillus perpallidus*; it was also significantly different from those of all other species. The lowest f0 min was found in pup USVs of *Sekeetamys calurus*; it differed significantly from those of all species except for *Meriones unguiculatus* (table 3).

The highest f0beg was found in pup USVs of *Dipodillus campestris*; it was significantly different from the f0beg of all other species. The lowest f0beg was found in pup USVs of *Meriones unguiculatus* (table 3), probably accounting for the increased prevalence of upward contour shapes in this species compared with others (figure 4). In other species, the values of f0beg were intermediate and did not differ significantly between species, except between *Gerbillus perpallidus* and *Sekeetamys calurus* (table 3).

The f0end was also highest in pup USVs of *Dipodillus campestris*; it was significantly different from the f0end of all other species. The lowest f0end was found in pup USVs of *Sekeetamys calurus*; it was significantly different from f0end of all other species. Pup *Gerbillus perpallidus* had a higher f0end, significantly different from the f0end of *Pachyuromys duprasi* and *Meriones vinogradovi* (table 3).

The highest fpeak was found in pup USVs of *Dipodillus campestris*; it was significantly different from the fpeak of all other species. The fpeak of pup USVs of *Gerbillus perpallidus* was also high but did not differ significantly from the fpeak of *Meriones vinogradovi*. The USVs of the other species were grouped together with species with low fpeak values (table 3 and figure 6).

Overall, the values of the f0 of pup USVs were always substantially and significantly higher in pup *Dipodillus campestris* (range approx. 65–75 kHz). Pup USVs of the other five species were in approximately the same f0 range (approx. 35–55 kHz).

GLMM showed that species identity affected all USV acoustic variables, while pup body size index (first PCA factor) only significantly affected the f0 min (table 3). Pup body size index significantly negatively correlated with f0 min ($r = -0.43$, $p < 0.001$, $N = 60$). Therefore, USV acoustics were mostly predetermined by species identity and only to a lesser extent by body size. Similar results were obtained for separate effects of log body weight, body length and head length (electronic supplementary material, table S2).

**Table 3.** Values (mean ± s.d.) of the acoustic variables of 6–10-day-old pup ultrasonic calls of six Gerbillinae species and GLMM results for species identity and body size index effects on the acoustic variables. Species was introduced as a categorical factor; body size index as a continuous factor. Designations: $N$, the number of ultrasonic calls, one 'averaged' call per pup, with the averaged values of each acoustic variable over 20 calls taken from each individual subject; duration, call duration; f0max, the maximum fundamental frequency; f0 min, the minimum fundamental frequency; f0beg, the fundamental frequency at the onset of a call; f0end, the fundamental frequency at the end of a call; fpeak, the frequency of maximum amplitude; D.c., *Dipodillus campestris*; G.p., *Gerbillus perpallidus*; M.u., *Meriones unguiculatus*; M.v., *Meriones vinogradovi*; S.c., *Sekeetamys calurus*; P.d., *Pachyuromys duprasi*. The same letters next to values indicate the lack of significant differences between them ($p > 0.05$, Tukey post hoc).

| acoustic variable | D.c., $N = 10$ | G.p., $N = 10$ | M.u., $N = 10$ | M.v., $N = 10$ | S.c., $N = 10$ | P.d., $N = 10$ | species identity | body size index |
|---|---|---|---|---|---|---|---|---|
| duration (ms) | $85.8 \pm 35.8^{a,c}$ | $82.5 \pm 22.6^{a,b}$ | $111.4 \pm 13.9^{c,d}$ | $158.7 \pm 10.8^{e}$ | $132.1 \pm 14.8^{d,e}$ | $56.2 \pm 19.7^{b}$ | $F_{5,53} = 25.15; p < 0.001$ | $F_{1,53} = 1.36; p = 0.25$ |
| f0max (kHz) | $74.8 \pm 3.7^{a}$ | $52.1 \pm 2.4^{b}$ | $50.0 \pm 3.5^{b,c}$ | $52.7 \pm 2.5^{b}$ | $47.8 \pm 1.8^{c}$ | $51.5 \pm 1.9^{b}$ | $F_{5,53} = 75.27; p < 0.001$ | $F_{1,53} = 2.92; p = 0.09$ |
| f0min (kHz) | $64.2 \pm 5.7^{a}$ | $47.2 \pm 3.3^{b}$ | $37.3 \pm 2.4^{c,d}$ | $40.0 \pm 2.4^{d}$ | $34.0 \pm 2.8^{d}$ | $40.2 \pm 3.5^{c}$ | $F_{5,53} = 113.30; p < 0.001$ | $F_{1,53} = 0.27; p = 0.60$ |
| f0beg (kHz) | $65.8 \pm 5.4^{a}$ | $48.4 \pm 3.4^{b}$ | $37.7 \pm 2.5^{c}$ | $45.1 \pm 2.7^{b,d}$ | $42.8 \pm 1.6^{d,e}$ | $45.6 \pm 2.9^{b,e}$ | $F_{5,53} = 78.71; p < 0.001$ | $F_{1,53} = 1.87; p = 0.18$ |
| f0end (kHz) | $67.4 \pm 4.9^{a}$ | $48.5 \pm 3.3^{b}$ | $45.9 \pm 3.7^{b,c}$ | $41.9 \pm 2.6^{c}$ | $34.7 \pm 2.7^{d}$ | $43.5 \pm 3.0^{c}$ | $F_{5,53} = 81.04; p < 0.001$ | $F_{1,53} = 5.20; p = 0.03$ |
| fpeak (kHz) | $71.9 \pm 3.6^{a}$ | $50.5 \pm 2.5^{b}$ | $44.5 \pm 2.1^{c}$ | $49.2 \pm 2.2^{b,d}$ | $45.3 \pm 1.8^{c}$ | $46.6 \pm 3.3^{c,d}$ | $F_{5,53} = 128.38; p < 0.001$ | $F_{1,53} = 0.04; p = 0.85$ |

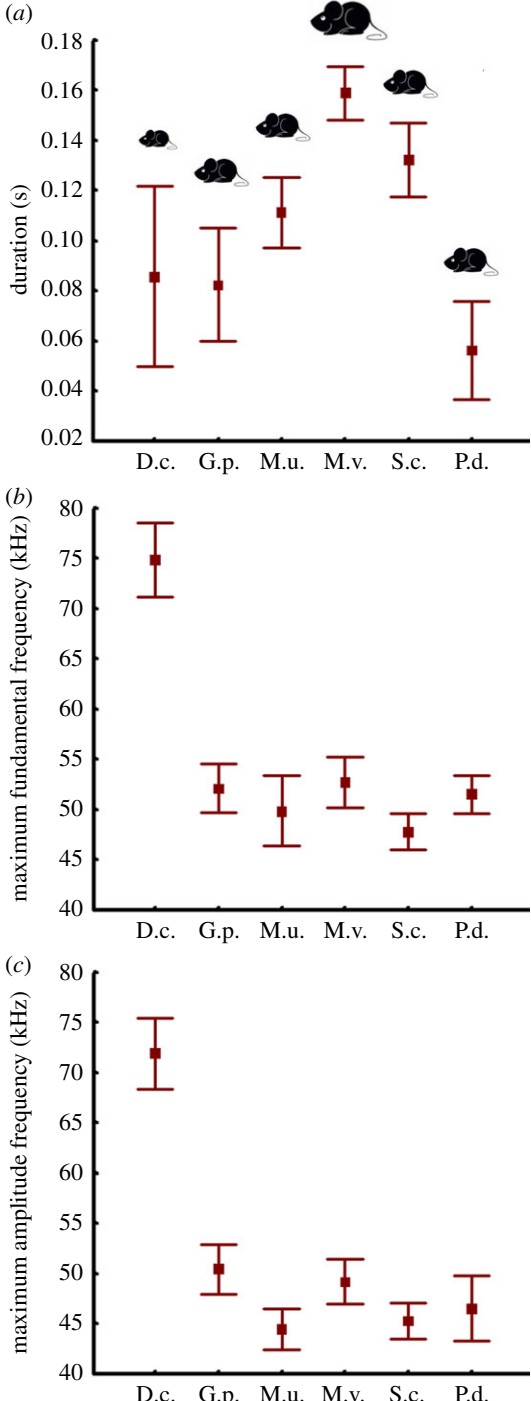

**Figure 6.** Values (mean ± s.d.) for (*a*) duration, (*b*) the maximum fundamental frequency (f0max) and (*c*) the maximum amplitude frequency (fpeak) of USV calls in 6–10-day-old pups of six Gerbillinae species. Central points indicate mean values, whiskers indicate s.d. Designations: D.c., *Dipodillus campestris*; G.p., *Gerbillus perpallidus*; M.u., *Meriones unguiculatus*; M.v., *Meriones vinogradovi*; S.c., *Sekeetamys calurus*; P.d., *Pachyuromys duprasi*. Pup body size is reflected in animal symbols.

## 3.4. Classifying calls to correct species with discriminant function analysis

The DFA, based on all six measured acoustic variables of USVs, assigned calls to species with an accuracy of 100% (figure 7). The accuracy of the DFA decreased to 98.3% when the more conservative leave-one-out cross-validation was applied. Only one single USV of *Meriones unguiculatus* was incorrectly assigned to *Gerbillus perpallidus*. Wilks' lambda values showed that the variables mainly contributing to discrimination were duration and f0beg (in order of decreasing importance) (table 4). The first

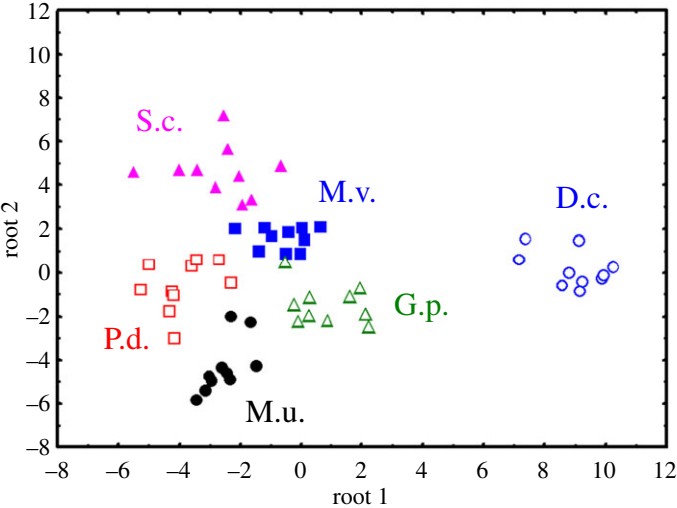

**Figure 7.** Scatterplot showing separation produced by the first two discriminant roots of 60 pups of six Gerbillinae species to correct species with DFA. DFA was based on the averaged values of acoustic variables of pup ultrasonic calls (one data point corresponds to one individual pup). Designations: D.c., *Dipodillus campestris*; G.p., *Gerbillus perpallidus*; M.u., *Meriones unguiculatus*; M.v., *Meriones vinogradovi*; S.c., *Sekeetamys calurus*; P.d., *Pachyuromys duprasi*.

**Table 4.** DFA results for ultrasonic calls of 6–10-day-old pups of six Gerbillinae species. Values of correlation between the acoustic variables of ultrasonic calls and the five discriminant functions, eigenvalues and per cent variances, described by each function, are given. The Wilks' lambda values and call variable effects are presented for each acoustic variable.

| acoustic variable | discriminant functions | | | | | Wilks' lambda | variable effect |
| --- | --- | --- | --- | --- | --- | --- | --- |
| | 1 | 2 | 3 | 4 | 5 | | |
| duration | −0.104 | −0.409 | 0.858 | 0.216 | −0.196 | 0.002164 | $F = 37.373$ |
| f0beg | 0.714 | −0.251 | −0.293 | −0.255 | −0.096 | 0.000938 | $F = 10.655$ |
| f0max | 0.800 | 0.112 | 0.087 | 0.492 | −0.270 | 0.000688 | $F = 5.192$ |
| f0end | 0.641 | 0.610 | 0.055 | −0.169 | −0.006 | 0.000757 | $F = 6.701$ |
| f0 min | 0.730 | 0.237 | −0.121 | −0.623 | −0.026 | 0.000716 | $F = 5.801$ |
| fpeak | 0.878 | −0.056 | 0.086 | 0.064 | 0.451 | 0.000666 | $F = 4.726$ |
| eigenvalue | 6.754 | 1.140 | 0.799 | 0.236 | 0.023 | | |
| per cent variance (%) | 75.45 | 12.73 | 8.93 | 2.64 | 0.25 | | |

discriminant function described 75.45% of the variance and was correlated with all fundamental frequency parameters (f0beg, f0max, f0end and f0 min) and peak frequency (table 4). The second discriminant function described 12.73% of the variance and correlated only with f0end. The third discriminant function described 8.93% of the variance and strongly correlated with duration. The fourth and fifth discriminant functions described only 2.89% of the variance (table 4).

## 4. Discussion

This comparative study revealed that 6–10-day-old pups of six Gerbillinae species were different in body size and in the acoustic traits of their isolation-induced ultrasonic calls. The species differed also in their most frequently used contour shapes of USVs. Nonlinear phenomena occurred mostly in *Pachyuromys duprasi* pups. The acoustic structure of the USVs was strictly species-specific, enabling 100% correct discrimination of calls by DFA.

Pup body size did not display any sustainable relationship with USV acoustic variables. Commonly, the larger-sized species produce longer and lower frequency calls because they have the larger sound-producing apparatus [95–97]. In partial support of this common rule, the longest USVs were found in the largest pups (*Meriones vinogradovi*) and the highest frequency USVs were found in the smallest pups (*Dipodillus campestris*) (table 3 and figure 6). However, across the six study species, the duration of USVs (ranging from 56–159 ms among species) did not display a significant relationship with pup body size; and, among frequency variables, only the minimum fundamental frequency of USVs depended on pup body size. The effect of species on the acoustics of USVs exceeded significantly the effect of body size (table 3). While pup gerbils produce both the ultrasonic and audible calls [52], further research should indicate whether the relationship between the acoustics and body size exists in the audible pup isolation-induced calls of Gerbillinae rodents.

The observed differences among species may be potentially explained by hypotheses of acoustic adaptation [98], social complexity [99,100], gene drift [101] and hearing ranges [102]. The acoustic adaptation hypothesis [98,103,104] suggests that pup USVs should be acoustically similar in gerbil species living in similar biotopes and acoustically different in species living in different biotopes. The species *Dipodillus campestris*, *Sekeetamys calurus* and *Meriones vinogradovi* all burrow in stony habitats [2,20,80,105], *Meriones unguiculatus* in arid steppes [20,106], and *Gerbillus perpallidus* and *Pachyuromys duprasi* both burrow in sands [2,107]. However, our data do not confirm this proposal, because the acoustic variables are not more similar in species burrowing in the same habitats (table 3). We can, therefore, conclude that there is no relationship between duration, fundamental frequency and peak frequency of pup USVs and biotope. On the other hand, the acoustic adaptation hypothesis was advanced for explaining the evolution of species-specific calls for better propagation in the environment. Infant pups of any gerbil species are still in burrows, so, distinctive to adult gerbils, the micro-environment in the burrow is practically the same, with relatively minor differences. Nevertheless, pup calls show clear species-specific differences in their acoustic features. The acoustic adaptation hypothesis was considered here for the first time in application to vocalizations of pup rodents, whereas previously it was only applied for explaining the evolution of species-specific vocalizations in adult rodents [108–111].

The social complexity hypothesis [100] suggests that within taxa, the differences in the complexity of vocal repertoire and sophistication of vocal communication can be related to the complexity of social relationships and to increase with the degree of species sociality [99,112,113]. In this study, we can estimate, for the first time, the complexity of the vocal repertoire of pup gerbils via the complexity of the acoustics of USVs (contour shape, percentage of vocal nonlinear phenomena). Nonlinear phenomena, complicating USV acoustic structure, were mostly presented in *Pachyuromys duprasi* (figure 5). Most variable contour shapes were found in *Dipodillus campestris* (figure 4), as neither contour occurred in over 50% of pup USVs in this species. In each other species, we found the species-characteristic contour, presented in over 50% of all calls (figure 4).

These six species are also different in sociality. *Dipodillus campestris* is solitary in the wild [2] and does not create social groups in captivity [20]. *Gerbillus perpallidus* is moderately social in captivity, where it can breed in cages in groups with 2–3 subsequent litters [20], but immediately shifts to solitary lifestyle upon being released to large outdoor enclosures [114]. *Meriones unguiculatus* is highly social in the wild [106,115,116], in semi-captive conditions [117,118] and in captivity [20]. *Meriones vinogradovi* is social in the wild [105]. Both *Meriones vinogradovi* and *Sekeetamys calurus* are highly social in captivity, as they can create social groups with more than one litter in a cage, whereas *Pachyuromys duprasi* is less social, as it only breeds in pairs ([20] and A.S.Z. 2018, unpublished data).

In contradiction to the social complexity hypothesis, the most complex pup USVs with most variable contour shapes were found in *Dipodillus campestris* (figure 4), which is the most solitary of the six study species. Consistently, USVs complicated by nonlinear phenomena were most frequent in the medium-social *Pachyuromys duprasi*. We, therefore, conclude that our results do not support the social complexity hypothesis.

The gene drift hypothesis suggests that the accumulation of non-specific mutations can be responsible for the acoustic differences in calls of closely related taxa [101,110,119]. Thus, the strongest acoustic differences can be found in most remotely related taxa. Among the six study gerbils, there are two pairs of closely related species, *Dipodillus campestris*–*Gerbillus perpallidus* and *Meriones unguiculatus*–*Meriones vinogradovi*, the genus *Sekeetamys* is related to the genus *Meriones* [120], while *Pachyuromys duprasi* is the most distant among the six species [120,121].

In agreement with gene drift hypothesis, *Pachyuromys duprasi* is distinctive qualitatively, because of the presence of numerous nonlinear phenomena in USVs (figure 5). Regarding the differences in the

acoustic variables, USVs of *Pachyuromys duprasi* are the shortest but do not differ significantly in duration from USVs of *Meriones unguiculatus*, *Sekeetamys calurus* or *Meriones vinogradovi* (table 3).

In contradiction to the gene drift hypothesis, USVs of two pairs of the closely related species *Dipodillus campestris*–*Gerbillus perpallidus* and *Meriones unguiculatus*–*Meriones vinogradovi* were similar to each other at the same degree as with calls of other study gerbil species. The USVs of *Dipodillus campestris* are the highest frequency, while USVs of all other study gerbil species are substantially lower frequency (figure 6). *Meriones unguiculatus* and *Meriones vinogradovi* distinctively use contour shapes: *Meriones unguiculatus* primarily uses the upward contour, while *Meriones vinogradovi* primarily uses the complex contour (figure 4). Call duration, the f0 at the onset of a call, and the peak frequency differ in *Meriones unguiculatus* and *Meriones vinogradovi*, although other variables are similar (table 3). We, therefore, conclude that the gene drift hypothesis is only partially supported by our results.

An early study for call-based taxonomy of *Pachyuromys duprasi* and five species of genus *Meriones* [25] also reports that similarities and dissimilarities of vocalizations (ultrasonic and audible) among these gerbil taxa are not sufficient for elucidating the relationship between the acoustics and phylogeny. However, more detailed data on the differences in the acoustics of USVs of adult gerbils of eight South African species support the results of chromosomal and molecular data [7,26,27].

The hearing range hypothesis suggests that the differences in the acoustics of USVs can be due to the differences in hearing ranges of adult animals, who are the recipients of pup isolation calls, e.g. see [102,122]. In *Pachyuromys duprasi*, hearing sensitivity is shifted towards low frequencies compared with species of the genus *Gerbillus* and *Meriones* [123–125]. This can be related to the morphological specialization in this species with strongly inflated eardrums and the enlarged malleus of the middle ear [123]. Based on this hypothesis, we could expect that the fundamental and peak frequencies of USVs of *Pachyuromys duprasi* should be lower than in gerbils belonging to the genus *Gerbillus* and *Meriones*. However, our results do not support this hypothesis, as USV fundamental frequencies are in the same range (table 3 and figure 6).

Ethics. This study was part of the research programme of the Scientific Research Department of Moscow Zoo. Two of the authors are zoo staff members, so no special permission was required for them to work with animals in Moscow Zoo. All study animals belonged to the laboratory collection of Moscow Zoo. The experimental procedure has been approved by the Committee of Bio-ethics of Lomonosov Moscow State University, research protocol no. 2011-36. We adhered to the 'Guidelines for the treatment of animals in behavioural research and teaching' [*Anim. Behav.*, 2020, **159**, I–XI] and to the laws on animal welfare for scientific research of the Russian Federation, where the study was conducted.

Data accessibility. All data are presented in the electronic supplementary material.

Authors' contributions. J.D.K. conducted the experiments, categorized vocalizations, conducted call measurements and drafted the manuscript. I.A.V. planned and carried out data collection, conducted the experiments, analysed the recordings, performed the statistical analysis, prepared figures and tables, guided the writing of the manuscript and provided revisions to the scientific content of the manuscript. A.S.Z. conducted the experiments, categorized vocalizations, conducted call measurements and drafted the manuscript. O.G.I. planned and carried out data collection, drafted the manuscript and provided revisions to the scientific content of the manuscript. E.V.V. conducted the experiments, analysed the recordings, drafted the manuscript, guided the writing of the manuscript and provided revisions to the scientific content of the manuscript.

Competing interests. We declare we have no competing interests.

Funding. This work was supported by the Russian Science Foundation (http://www.rscf.ru/) funding to J.D.K., I.A.V., A.S.Z. and E.V.V. (grant no. 19-14-00037).

Acknowledgements. We are sincerely grateful to Elena Neprintseva for her help with organizing the experiments for this study, to Olga Filatova for her help with statistics and to the staff of the Department of Small Mammals of Moscow Zoo, for their permanent help and support. We thank the two anonymous reviewers for their detailed and constructive comments.

Disclaimer. The funder had no role in study design, data collection and analysis, decision to publish or preparation of the manuscript.

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
