## [Peer Review File · Royal Society Open Science]

Review History

RSOS-201558.R0 (Original submission)

Review form: Reviewer 1

Is the manuscript scientifically sound in its present form?

Yes

Are the interpretations and conclusions justified by the results?

Yes

Is the language acceptable?

Yes

Do you have any ethical concerns with this paper?

No

Have you any concerns about statistical analyses in this paper?

Yes

Recommendation?

Accept with minor revision (please list in comments)

Comments to the Author(s)

See attached file (Appendix A).

Review form: Reviewer 2**Is the manuscript scientifically sound in its present form?**

No

Are the interpretations and conclusions justified by the results?

No

Is the language acceptable?

No

Do you have any ethical concerns with this paper?

No

Have you any concerns about statistical analyses in this paper?

Yes

Recommendation?

Major revision is needed (please make suggestions in comments)

Comments to the Author(s)

You have presented a novel and interesting study on the species differences in infant Gerbil isolation calls. Extensive data collection and acoustical analyses have been undertaken to characterise these calls, so you should be commended. I have some queries and suggestions over the statistical analyses and interpretation of results.

I would additionally suggest to have the English of this paper edited.

More specifically:

- In the abstract, please mention what vocal variables were measured. E.g. pitch, temporal parameters etc.

-Line 49 - please clarify the statement starting with "the effect of species identity..."

- If you report on average values within the manuscript, you need to specify this in your abstract, as it looks like you have extracted this information from the entire 1200 calls rather than collapsing it down.

Introduction: All in all, this section could be written more concisely. A lot of basic background information was provided on the species, and further synthesis is needed. The gap in knowledge also needs to be more distinctly highlighted.

- Please correct statement to "Gerbils or jirds are"

Materials and methods:

- Some of the information provided on the gerbil test subjects may be better presented in tabular form. E.g. species, number of subjects, origin, year of data collection, age of subjects.

- How did you determine which pups to select in each litter? Was it randomized or balanced for sex?
- Line 164 – with minimal reverberation.
- Line 177 – to minimise the manipulations on the pup during the trial.
- Line 191 does not make sense. Successively would be a better word than consequently.
- Line 199 – 204 – sentence a bit hard to follow. Please rephrase.
- Do you know how many calls in total were there to select from?
- Line 226 – according to.
- Please confirm - Was it a single observer conducting the classification of the f0 contour shapes?
- Statistical analyses – there are updated techniques for incorporating repeated measures into your statistical analyses. Given that your dataset is of adequate size, I would recommend using one of these approaches rather than taking the average for individual subjects. Instead of Anova you could try a linear mixed effect model, accounting for individual pup as a random effect.
- Further, seeing as you have collected 1200 calls, it would be interesting to see the DFA classification percentages when the entire dataset is used. See Favaro et al., 2016. Favaro, L.; Gili, C.; Da Rugna, C.; Gnone, G.; Fissore, C.; Sanchez, D.; McElligott, A. G.; Gamba, M.; Pessani, D. Vocal Individuality and Species Divergence in the Contact Calls of Banded Penguins. *Behav. Processes* 2016, 128, 83–88. <https://doi.org/10.1016/j.beproc.2016.04.010>. I think this approach would be more appropriate and would reduce your classification accuracy (which is extremely high).
- For your current GLMMs, what was the family that you used? Further, please specify your random effects.
- Please mention the number of acoustic variables measured in the stat analysis.
- For the DFA, was a cross- validation procedure undertaken?
- I would suggest conducting a Manova as well.

Results:

- Line 298 - present not usual.
- Body size PCA -I would also highlight the high eigenvalue of the first PC compared to the other two.
- DFA - mention which variables are loaded onto which discriminant functions.
- Figure 5 – it is difficult to see the percentages of biphonation and frequency jumps for some species. Could you instead graph the proportion of biphonation and fj in the species where NLP is present?
- Table S2 - continuous factors not continual.
- Have you considered classifying the calls to the correct individual within each species? i.e. conducting 6 separate DFAs. This would help confirm or refute your species classification percentages.

Discussion:

- You mention the presence of NLP and the qualitative distinctiveness of *pachyuromys duprasi*. But I cannot see any inferential stats for the NLP and USV contour shapes. Could you not do a binomial GLMM to compare the presence or absence of NLP between species? Further, a multinomial model to compare the usv shapes amongst the species? This would further support your claims.
- It is a bit hard to keep track of all the hypotheses, especially when your results do not align with some of them. I'd suggest focusing on the more relevant and relatable hypotheses.

Decision letter (RSOS-201558.R0)

Dear Dr Volodin

The Editors assigned to your paper RSOS-201558 "Pup ultrasonic isolation calls of six gerbil species and the relationship between acoustic traits and body size" have now received comments from reviewers and would like you to revise the paper in accordance with the reviewer comments and any comments from the Editors. Please note this decision does not guarantee eventual acceptance.

Please submit your revised manuscript and required files (see below) no later than 21 days from today's (ie 11-Dec-2020) date. Note: the ScholarOne system will 'lock' if submission of the revision is attempted 21 or more days after the deadline. If you do not think you will be able to meet this deadline please contact the editorial office immediately.

on behalf of Dr Claudia Wascher (Associate Editor) and Kevin Padian (Subject Editor)
openscience@royalsociety.org

Associate Editor Comments to Author (Dr Claudia Wascher):

Two reviewers have now commented on this manuscript, which investigates the relationship between ultrasonic calls features and body size in pups of six gerbil species. Both reviewers find the presented study interesting and generally well presented. The reviewers recommend

clarification of the general framework and re-analysis of the data before publication of the manuscript.

Reviewer comments to Author:

Reviewer: 1

Comments to the Author(s)

See attached file.

Reviewer: 2

Comments to the Author(s)

You have presented a novel and interesting study on the species differences in infant Gerbil isolation calls. Extensive data collection and acoustical analyses have been undertaken to characterise these calls, so you should be commended. I have some queries and suggestions over the statistical analyses and interpretation of results.

I would additionally suggest to have the English of this paper edited.

More specifically:

- In the abstract, please mention what vocal variables were measured. E.g. pitch, temporal parameters etc.

- Line 49 - please clarify the statement starting with "the effect of species identity..."

- If you report on average values within the manuscript, you need to specify this in your abstract, as it looks like you have extracted this information from the entire 1200 calls rather than collapsing it down.

Introduction: All in all, this section could be written more concisely. A lot of basic background information was provided on the species, and further synthesis is needed. The gap in knowledge also needs to be more distinctly highlighted.

- Please correct statement to "Gerbils or jirds are"

Materials and methods:

- Some of the information provided on the gerbil test subjects may be better presented in tabular form. E.g. species, number of subjects, origin, year of data collection, age of subjects.

- How did you determine which pups to select in each litter? Was it randomized or balanced for sex?

- Line 164 - with minimal reverberation.

- Line 177 - to minimise the manipulations on the pup during the trial.

- Line 191 does not make sense. Successively would be a better word than consequently.

- Line 199 - 204 - sentence a bit hard to follow. Please rephrase.

- Do you know how many calls in total were there to select from?

- Line 226 - according to.

- Please confirm - Was it a single observer conducting the classification of the f0 contour shapes?

Statistical analyses - there are updated techniques for incorporating repeated measures into your statistical analyses. Given that your dataset is of adequate size, I would recommend using one of these approaches rather than taking the average for individual subjects. Instead of Anova you could try a linear mixed effect model, accounting for individual pup as a random effect.

- Further, seeing as you have collected 1200 calls, it would be interesting to see the DFA classification percentages when the entire dataset is used. See Favaro et al., 2016. Favaro, L.; Gili, C.; Da Rugna, C.; Gnone, G.; Fissore, C.; Sanchez, D.; McElligott, A. G.; Gamba, M.; Pessani, D. Vocal Individuality and Species Divergence in the Contact Calls of Banded Penguins. *Behav. Processes* 2016, 128, 83-88. <https://doi.org/10.1016/j.beproc.2016.04.010>. I think this approach

would be more appropriate and would reduce your classification accuracy (which is extremely high).

- For your current GLMMs, what was the family that you used? Further, please specify your random effects.
- Please mention the number of acoustic variables measured in the stat analysis.
- For the DFA, was a cross-validation procedure undertaken?
- I would suggest conducting a Manova as well.

Results:

- Line 298 - present not usual.
- Body size PCA - I would also highlight the high eigenvalue of the first PC compared to the other two.
- DFA - mention which variables are loaded onto which discriminant functions.
- Figure 5 - it is difficult to see the percentages of biphonation and frequency jumps for some species. Could you instead graph the proportion of biphonation and fj in the species where NLP is present?
- Table S2 - continuous factors not continual.
- Have you considered classifying the calls to the correct individual within each species? i.e. conducting 6 separate DFAs. This would help confirm or refute your species classification percentages.

Discussion:

- You mention the presence of NLP and the qualitative distinctiveness of *pachyuromys duprasi*. But I cannot see any inferential stats for the NLP and USV contour shapes. Could you not do a binomial GLMM to compare the presence or absence of NLP between species? Further, a multinomial model to compare the usv shapes amongst the species? This would further support your claims.
- It is a bit hard to keep track of all the hypotheses, especially when your results do not align with some of them. I'd suggest focusing on the more relevant and relatable hypotheses.

===PREPARING YOUR MANUSCRIPT===

===PREPARING YOUR REVISION IN SCHOLARONE===

Author's Response to Decision Letter for (RSOS-201558.R0)

See Appendix B.

RSOS-201558.R1 (Revision)

Review form: Reviewer 1

Is the manuscript scientifically sound in its present form?

Yes

Are the interpretations and conclusions justified by the results?

Yes

Is the language acceptable?

Yes

Do you have any ethical concerns with this paper?

No

Have you any concerns about statistical analyses in this paper?

No

Recommendation?

Accept as is

Comments to the Author(s)

Dear Authors, I reiterate my appreciation for your study, which I believe interesting and properly conducted. The concerns regarding the general framework of the work and the discussion general economy have been dispelled. Moreover, when comments or suggestions have been refused, I found your answers and explanation perfectly reasonable.

Review form: Reviewer 2

Is the manuscript scientifically sound in its present form?

Yes

Are the interpretations and conclusions justified by the results?

Yes

Is the language acceptable?

No

Do you have any ethical concerns with this paper?

No

Have you any concerns about statistical analyses in this paper?

Yes

Recommendation?

Accept with minor revision (please list in comments)

Comments to the Author(s)

Thank you for reviewing my manuscript suggestions and providing detailed answers to them. I still have some minor queries about the statistical approach, which once answered, I will be happy to accept the manuscript. Please see below:

Line 124: Some typos - please amend.

Line 251: You previously mentioned that you log transformed the body weight variable. This should also be mentioned here - that you log transformed to satisfy this assumption of normality.

Line 254: Tukey's Honest Significant Difference.

Line 254: There is plenty of literature which argues that the mixed method approach is more appropriate for dealing with repeated measures and avoiding type 1 errors associated with pseudoreplication. Please clarify/correct paragraph starting on line 254 as this part is still unclear. Please justify choice of one-way ANOVA with averaged values, as opposed to a mixed-effect model with the full dataset, accounting for repeated measures using individual ID as a random effect. You mention in your response that you conducted the LMM, so these should be reported, along with the estimated marginal means for each species.

Line 258/comment 49: You have mentioned that you used a GLMM, but you have not mentioned what the random effect in this model is. Please clarify. Did you include individual identity as a random effect? Were species and body size included in the same model, with acoustic parameter as the outcome? Accordingly, I am still confused by some of the results, e.g. Table 4. Why not conduct post-hoc comparisons for the GLMM instead of the ANOVA in table 3?

Line 260: Specify response and categorical factors for the binomial GLM.

Line 301/comment 53: Usual still does not really make sense here. Common a better phrase.

Line 348: Correlated significantly negatively a bit awkward. Rephrase.

Line 356: Typo. Single

Table 5: Rename root as discriminant function 1, df2... etc.

Decision letter (RSOS-201558.R1)

Dear Dr Volodin

On behalf of the Editors, we are pleased to inform you that your Manuscript RSOS-201558.R1 "Pup ultrasonic isolation calls of six gerbil species and the relationship between acoustic traits and body size" has been accepted for publication in Royal Society Open Science subject to minor revision in accordance with the referees' reports. Please find the referees' comments along with any feedback from the Editors below my signature.

Please submit your revised manuscript and required files (see below) no later than 7 days from today's (ie 03-Feb-2021) date. Note: the ScholarOne system will 'lock' if submission of the revision is attempted 7 or more days after the deadline. If you do not think you will be able to meet this deadline please contact the editorial office immediately.

on behalf of Dr Claudia Wascher (Associate Editor) and Kevin Padian (Subject Editor)
openscience@royalsociety.org

Editor comments:

Thank you for your attention to the reviewers' comments in your resubmission. One reviewer is now happy with the manuscript, and the other has some remaining concerns that need to be addressed specifically in your final revision. We look forward to your corrected manuscript.

Reviewer comments to Author:

Reviewer: 2

Comments to the Author(s)

Thank you for reviewing my manuscript suggestions and providing detailed answers to them. I still have some minor queries about the statistical approach, which once answered, I will be happy to accept the manuscript. Please see below:

Line 124: Some typos - please amend.

Line 251: You previously mentioned that you log transformed the body weight variable. This should also be mentioned here - that you log transformed to satisfy this assumption of normality.

Line 254: Tukey's Honest Significant Difference.

Line 254: There is plenty of literature which argues that the mixed method approach is more appropriate for dealing with repeated measures and avoiding type 1 errors associated with pseudoreplication. Please clarify/correct paragraph starting on line 254 as this part is still unclear. Please justify choice of one-way ANOVA with averaged values, as opposed to a mixed-effect model with the full dataset, accounting for repeated measures using individual ID as a random effect. You mention in your response that you conducted the LMM, so these should be reported, along with the estimated marginal means for each species.

Line 258/comment 49: You have mentioned that you used a GLMM, but you have not mentioned what the random effect in this model is. Please clarify. Did you include individual identity as a random effect? Were species and body size included in the same model, with acoustic parameter as the outcome? Accordingly, I am still confused by some of the results, e.g. Table 4. Why not conduct post-hoc comparisons for the GLMM instead of the ANOVA in table 3?

Line 260: Specify response and categorical factors for the binomial GLM.

Line 301/comment 53: Usual still does not really make sense here. Common a better phrase.

Line 348: Correlated significantly negatively a bit awkward. Rephrase.

Line 356: Typo. Single

Table 5: Rename root as discriminant function 1, df2... etc.

Reviewer: 1

Comments to the Author(s)

Dear Authors, I reiterate my appreciation for your study, which I believe interesting and properly conducted. The concerns regarding the general framework of the work and the discussion general economy have been dispelled. Moreover, when comments or suggestions have been refused, I found your answers and explanation perfectly reasonable.

===PREPARING YOUR MANUSCRIPT===

===PREPARING YOUR REVISION IN SCHOLARONE===

Author's Response to Decision Letter for (RSOS-201558.R1)

See Appendix C.

Decision letter (RSOS-201558.R2)

Dear Dr Volodin,

It is a pleasure to accept your manuscript entitled "Pup ultrasonic isolation calls of six gerbil species and the relationship between acoustic traits and body size" in its current form for publication in Royal Society Open Science.

Please see the Royal Society Publishing guidance on how you may share your accepted author manuscript at <https://royalsociety.org/journals/ethics-policies/media-embargo/>. After publication, some additional ways to effectively promote your article can also be found here

<https://royalsociety.org/blog/2020/07/promoting-your-latest-paper-and-tracking-your-results/>.

on behalf of Dr Claudia Wascher (Associate Editor) and Kevin Padian (Subject Editor)
openscience@royalsociety.org

Appendix A

RSOS-201558

Pup ultrasonic isolation calls of six gerbil species and the relationship between acoustic traits and body size.

The manuscript investigates the relationship between ultrasonic calls features and body size in pups of six gerbil species. I deem the manuscript concise and yet thorough. The data collection as well as the acoustic analyses appear appropriate for testing the authors' hypothesis and I have no concerns regarding ethics and research integrity, clearly stated. The authors shared the data, therefore I have no concerns regarding their availability and transparency either. Still, I have some doubts about the general framework and interpretation, as well as some concerns regarding the statistical analyses. If these can be fixed, then this should become a valuable contribution to the literature.

1) Regarding the general framework, the authors stated that the results are discussed 'with the hypotheses of acoustic adaptation, social complexity, hearing ranges, the effect of body size and phylogeny', and indeed the discussion considered those major hypotheses, often contemplated in these kinds of investigation. My concern arises because, while the rationale and the methodology supporting the body size hypothesis are thoroughly elucidated, the other theories are only considered in discussion, and are not reported anywhere else (most of the introduction [LL83-114] focuses on the differences in body mass). Moreover, despite the core of the investigation is the relationship between body size and acoustic traits, in the discussion this is just one of the points considered (and the last, no less). To recap, I have no concerns regarding the discussion of the results with the stated theories, but it would be nice to construct an introductory framework that includes them. I also think necessary to reassess the discussion general economy, especially considering that the investigation regards the body size hypothesis only.

2) Regarding the interpretation of the results with the social complexity hypothesis, I understand that the call types classification derives from previous research, already published. Still, I am afraid that classifying a call as 'complex' could be confusing, when calls considered complex with regards to the social complexity hypothesis are those calls including frequency jumps and biphonation, two features not strictly related to the USV call with complex contour. I, therefore, think that can be worth evaluating an alternative to the term complex (i.e. *vibrato-like call*, see 'Function and Evolution of Vibrato-like Frequency Modulation in Mammals' Charlton et al. 2017 'Current Biology'). However, I would like to reiterate that this is just a suggestion.

3) Considering the statistical analyses, I did not understand how the authors built the GLMM, nor its results (see following comments).

Below more punctual comments, for each section of the draft.

STATEMENT

Maybe 'Two of the authors' instead of 'The two authors'?

INTRODUCTION

L75 I imagine 'USV' probably stands for 'ultrasonic vocalisations', but better be clear, at least the first time. Moreover, if it is so, ultrasonic vocalisations calls would be redundant (throughout the manuscript).

METHODS

LL125-134 I think this part could be shortened.

L164 'High signal/noise ratio'. Can the authors be more specific?

L202 'eligible'. How is the eligibility defined?

L208 '20 USV calls per pup'. Already stated at L200.

L259 I really do not understand how the GLMM is built nor its results. Which is the response variable? From Table4/S2 Table it seems that the acoustic variables are considered as response variable. But, if so, how can it be possible to use just one model? Can the authors report the formula, as well as the complete results?

LL259-261 The authors stated: 'We used Generalized Linear Mixed Model (GLMM) for estimating 'the joint effect of pup species and pup body size on the acoustic variables of USV calls', with species identity as a categorical factor and index body size as a continuous factor'. Does the 'joint effect' mean that the authors considered an interaction between these two factors? I do not really understand.

RESULTS

Overall, I think the species' scientific name can be shortened (at least the genus: i.e. *G. perpallidus* instead of *Gerbillus perpallidus*).

L292 I do not agree with the use of 'preferred'.

LL293-294 Would the authors consider of using 'flat contour' instead of 'contour flat' (same for upward and complex).

L306 I think that adding a figure reporting which are the call types showing nonlinear phenomena (rather than the species only) could be helpful.

LL340-342 'Pup USV calls of the other five species were in approximately the same f0 range (about 35-55 kHz); the differences in f0 values between them were in the details'. I agree that what the f0 range is can be easily deduced, but it is not reported in the methods. Moreover, what does 'in the details' mean?

L345 'Pup body size correlated significantly negatively with f0min ($r=-0.43$, $p<0.001$, $N=60$)'. Would the authors consider to rephrase it? 'Pup body size resulted negatively correlated with f0min ($r=-0.43$, $p<0.001$, $N=60$)' can be a possibility.

L347 'log body weight' is not reported in the methods.

DISCUSSION

L359 As already stated in the comment at L292, I do not agree with the use of 'preferential'.

L363 I would use 'among' instead of 'between'.

LL364-365 I have no concerns about the discussion of the results with regards to the stated theories, but it would be nice to see something about them within the introduction, too. Especially considering that the investigation really regards only one of those theories.

LL378-379 Would the authors consider to rephrase 'Nevertheless, pup calls have clear species-specific differences in the acoustics' in 'Nevertheless, pup calls show clear species-specific differences in their acoustic features'?

LL379-380 The relationship between this sentence and the previous one is unclear.

L382 Missing comma after 'elaboration' but the whole sentence at LL381-383 is not very clear. Please, consider to rephrase it.

LL384-385 'We can consider this hypothesis in application to our data, in spite of difficulties with estimating the complexity of the vocal repertoire in pup gerbils'. Can the authors add a reference?

LL386-387 We can estimate the complexity of the vocal repertoire via the complexity of the acoustics of USV calls (contour shape, percentage of vocal nonlinear phenomena). As in the previous comment, can the authors add a reference?

LL402-403 'In contradiction to the social complexity hypothesis, the most complex pup USV calls with most variable contour shapes were found in *Dipodillus campestris* (Fig 4)...' I am not really sure about referring to figures (or tables), within the discussion (see also L423, L426, L427...)

LL404-4045 'Consistently, USV calls complicating with nonlinear phenomena, were mostly frequent in the medium-social *Pachyuromys duprasi*.' The whole sentence is a bit unclear. 'Consistently' with the same hypothesis? Maybe 'complicated by' instead of 'complicating with'?

L416 Not really sure about using 'quantitative differences'

LL426 'Call duration, $f_{0\text{beg}}$ and f_{peak} differ in *Meriones unguiculatus*...' Missing comma after $f_{0\text{beg}}$; moreover, I would use the extended names (i.e. initial f_0).

LL429-432 'The early study for call-based taxonomy of *Pachyuromys duprasi* and five species of genus *Meriones* [25] also reports that similarities and dissimilarities of vocalizations (ultrasonic and audible) between these gerbil taxa are not sufficient for any reliable conclusions about the relationship between the acoustics and phylogeny'. Maybe 'An early study' instead of 'The early study'; 'among' instead of 'between'. However, would the author consider to rephrase the whole sentence?

LL453-455 'The effect of species-specific characteristics on the acoustics of USV calls exceeded significantly the effect of body size'. What do the authors mean with 'species-specific characteristics'?

LL456-462 Do not really understand why the conclusion addresses the audible calls, all of a sudden. It can surely represent a further direction but I would prefer a conclusion about the USV calls rather than the audible ones.

Appendix B

Dear Dr Claudia Wascher,

Dear Dr Kevin Padian,

We revised the MS ID: RSOS-201558 Title: Pup ultrasonic isolation calls of six gerbil species and the relationship between acoustic traits and body size.

We addressed most comments of the reviewers and provided responses to the comments below. According to the comments, we corrected the MS style and grammar. According to suggestions by Reviewer 1, we re-structured the Discussion, with accent on the effects of body size (supported by our results) among the other explaining hypotheses. As was suggested by Reviewer 1, we deleted from Discussion the paragraph about the audible calls. We provided additional statistical analyses suggested by Reviewer 2. We added Table 5 with DFA results for values of correlation between acoustic variable of ultrasonic calls and the five discriminant functions (roots), eigenvalues and percent variances. We did not include the analyses of vocal individual identity in the MS, because the aim of this study was in the comparison of pup ultrasonic isolation calls among six gerbil species. We plan to make a comparison of vocal individuality in pup ultrasonic calls in a separate study.

All changes in MS are labeled in green. All coauthors approved the submission of the revised version.

Yours sincerely,

Ilya Volodin, corresponding author

Dr. Ilya A. Volodin

Lab. of Animal Behaviour,

Dept. of Vertebrate Zoology,

Faculty of Biology,

Lomonosov Moscow State University,

Vorobiev Gory, 1/12, Moscow,

119234, Russia

Tel. +7 (495) 939-4578

e-mail volodinsvoc@gmail.com

e-mail volodinsvoc@mail.ru

<http://www.bioacoustica.org/>

Associate Editor Comments to Author (Dr Claudia Wascher):

Two reviewers have now commented on this manuscript, which investigates the relationship between ultrasonic calls features and body size in pups of six gerbil species. Both reviewers find the presented study interesting and generally well presented. The reviewers recommend clarification of the general framework and re-analysis of the data before publication of the manuscript.

Reviewer comments to Author:

Reviewer: 1

The manuscript investigates the relationship between ultrasonic calls features and body size in pups of six gerbil species. I deem the manuscript concise and yet thorough. The data collection as well as the acoustic analyses appear appropriate for testing the authors' hypothesis and I have no concerns regarding ethics and research integrity, clearly stated. The authors shared the data, therefore I have no concerns regarding their availability and transparency either. Still, I have some doubts about the general framework and interpretation, as well as some concerns regarding the statistical analyses. If these can be fixed, then this should become a valuable contribution to the literature.

Comment 1

Regarding the general framework, the authors stated that the results are discussed 'with the hypotheses of acoustic adaptation, social complexity, hearing ranges, the effect of body size and phylogeny', and indeed the discussion considered those major hypotheses, often contemplated in these kinds of investigation. My concern arises because, while the rationale and the methodology supporting the body size hypothesis are thoroughly elucidated, the other theories are only considered in discussion, and are not reported anywhere else (most of the introduction [LL83-114] focuses on the differences in body mass). Moreover, despite the core of the investigation is the relationship between body size and acoustic traits, in the discussion this is just one of the points considered (and the last, no less). To recap, I have no concerns regarding the discussion of the results with the stated theories, but it would be nice to construct an introductory framework that includes them. I also think necessary to reassess the discussion general economy, especially considering that the investigation regards the body size hypothesis only.

Response 1

We re-structured the Discussion, with accent on the paragraph on the effects of body size (supported by our results) among the other explaining hypotheses. As was suggested by Reviewer 1, we deleted from Discussion the paragraph about the audible calls.

Comment 2

Regarding the interpretation of the results with the social complexity hypothesis, I understand that the call types classification derives from previous research, already published. Still, I am afraid that classifying a call as 'complex' could be confusing, when calls considered complex with regards to the social complexity hypothesis are those calls including frequency jumps and biphonation, two features not strictly related to the USV call with complex contour. I, therefore, think that can be worth evaluating an alternative to the term complex (i.e. vibrato-like call, see 'Function and Evolution of Vibrato-like Frequency Modulation in Mammals' Charlton et al. 2017 'Current Biology'). However, I would like to reiterate that this is just a suggestion.

Response 2

Creating a new term "vibrato-like calls" for the ultrasonic vocalizations is undesirable, because it is inconsistent with previous studies and the literature on the ultrasonic vocalization in rodents and therefore can be confusing for the readers. The vibrato-like frequency modulation, accenting the vocal tract resonances (formants) is produced during phonation with vibration of the vocal folds in the larynx (Charlton et al. 2017). At the same time, our study considers the ultrasonic calls, produced by the whistling mechanism in the vocal tract and does not highlighting the formants. The term "complex" is very commonly applied for the ultrasonic call contours in many rodent species (including a broad literature on laboratory mice). This term "complex" might not be confusing for the readers working in the field of the audible vocalizations if the illustrative spectrograms and adequate descriptions are provided (as was done in this MS) and is applied here for consistency with previous research.

Comment 3

Considering the statistical analyses, I did not understand how the authors built the GLMM, nor its results (see following comments).

Response 3

Please see Responses 10 and 11 below regarding the changes in the statistical analyses.

Below more punctual comments, for each section of the draft.

STATEMENT

Comment 4

Maybe 'Two of the authors' instead of 'The two authors'?

Response 4

Was corrected as recommended (Line 465)

INTRODUCTION

Comment 5

L75 I imagine 'USV' probably stands for 'ultrasonic vocalisations', but better be clear, at least the first time. Moreover, if it is so, ultrasonic vocalisations calls would be redundant (throughout the manuscript).

Response 5

Following this recommendation, we decoded USV as “ultrasonic vocalizations (USVs)” at first mention in Introduction (Line 73) and deleted the redundant words “calls” after USV throughout the MS.

METHODS

Comment 6

LL125-134 I think this part could be shortened.

Response 6

We shortened this part as follows (Lines 123-130): "The *Dipodillus campestris* colony originated in 1989 from four individuals from the Museum of Natural History, France; the *Gerbillus perpallidus* colony originated in 1985 from three individuals obtained from German zoos; the *Meriones unguiculatus* colony originated in 2009 from 11 individuals from a natural colony in Tuva, Russia; the *Meriones vinogradovi* colony originated in 2006 from a natural colony (Armenia); the *Sekeetamys calurus* colony originated in 2002 from four individuals from zoos (Germany) and four individuals obtained in 2009 from a natural colony in United Arab Emirates; *Pachyuromys duprasi* colony originated in 2007 from eight individuals from Egypt."

Comment 7

L164 'High signal/noise ratio'. Can the authors be more specific?

Response 7

According to this comment and the suggestions by Reviewer 2, we re-wrote this sentence as follows (Lines 160-162): “The obtained recordings had a high signal/noise ratio, with calls not masked with background noise and minimal reverberation.”

Comment 8

L202 'eligible'. How is the eligibility defined?

Response 8

We mean calls of high-quality, not superimposed with noises. We deleted “eligible” from the text and re-wrote the sentence as follows (Lines 196-201): “Using visual inspection of spectrograms of acoustic files created with Avisoft SASLab Pro software (Avisoft Bioacoustics, Berlin, Germany) we selected 20 USVs per individual, taking calls randomly from those with high signal-to-noise ratio and without superimposed noise from different parts of each 120 s recording, approximately one ultrasonic call per 5-6 s, avoiding taking calls following each other.”

Comment 9

L208 '20 USV calls per pup'. Already stated at L200.

Response 9

We deleted '20 USV calls per pup' to avoid the redundancy.

Comment 10

L259 I really do not understand how the GLMM is built nor its results. Which is the response variable? From Table4/S2 Table it seems that the acoustic variables are considered as response variable. But, if so, how can it be possible to use just one model? Can the authors report the formula, as well as the complete results?

Response 10

The acoustic variables are considered as response (dependent) variables. The GLMM design in STATISTICA software enables introducing a few independent variants in the analysis simultaneously.

Comment 11

LL259-261 The authors stated: 'We used Generalized Linear Mixed Model (GLMM) for estimating 'the joint effect of pup species and pup body size on the acoustic variables of USV calls', with species identity as a categorical factor and index body size as a continuous factor'. Does the 'joint effect' mean that the authors considered an interaction between these two factors? I do not really understand.

Response 11

Thank you, we corrected this mistake and re-wrote the text as follows (Lines 256-258): "We used Generalized Linear Mixed Model (GLMM) for estimating the effects of pup species and pup body size on the acoustic variables of USVs, with pup species as a categorical factor and index body size as a continuous factor."

RESULTS

Comment 12

Overall, I think the species' scientific name can be shortened (at least the genus: i.e. *G. perpallidus* instead of *Gerbillus perpallidus*).

Response 12

Shortening the scientific names will result in the confusions (of e.g. between genus *Gerbillus* and genus *Gerbillurus*), because, in addition to the six study species belonging to the five genera, we mention/discuss many other species and genera of gerbils in this MS. Retaining the full names avoids the confusions.

Comment 13

L292 I do not agree with the use of 'preferred'.

Response 13

We re-wrote this sentence as follows (Lines 292-293): "Species differed substantially in their most frequently used contour shape of USVs."

Comment 14

LL293-294 Would the authors consider of using 'flat contour' instead of 'contour flat' (same for upward and complex).

Response 14

Was corrected as recommended (Lines 293-297).

Comment 15

L306 I think that adding a figure reporting which are the call types showing nonlinear phenomena (rather than the species only) could be helpful.

Response 15

The Figure 3 with call types showing nonlinear phenomena is already presented.

Comment 16

LL340-342 'Pup USV calls of the other five species were in approximately the same f_0 range (about 35-55 kHz); the differences in f_0 values between them were in the details'. I agree that what the f_0 range is can be easily deduced, but it is not reported in the methods. Moreover, what does 'in the details' mean?

Response 16

We deleted "the differences in f0 values between them were in the details". In the section "Acoustic analysis", we describe in detail how we measured f0 (Lines 209-220). In addition, measurements of all f0-related variables are illustrated on the Figure 1.

Comment 17

L345 'Pup body size correlated significantly negatively with f0min ($r=-0.43$, $p<0.001$, $N=60$)'. Would the authors consider to rephrase it? 'Pup body size resulted negatively correlated with f0min ($r=-0.43$, $p<0.001$, $N=60$)' can be a possibility.

Response 17

The sentence 'Pup body size correlated significantly negatively with f0min ($r=-0.43$, $p<0.001$, $N=60$)' was copied incorrectly from the MS text. The original sentence is (Lines 345-346): "Pup body size index correlated significantly negatively with f0min ($r=-0.43$, $p<0.001$, $N=60$)."

Comment 18

L347 'log body weight' is not reported in the methods.

Response 18

We added in the Methods (183-185): "The body variables and body weight (or log body weight) were taken as proxies of body size for further comparison with the USV acoustic variables."

DISCUSSION

Comment 19

L359 As already stated in the comment at L292, I do not agree with the use of 'preferential'.

Response 19

We re-wrote this sentence as follows (Lines 366-367): "The species differed also in their most frequently used contour shapes of USVs."

Comment 20

L363 I would use 'among' instead of 'between'.

Response 20

We replaced "between" with "among", as recommended (Line 382).

Comment 21

LL364-365 I have no concerns about the discussion of the results with regards to the stated theories, but it would be nice to see something about them within the introduction, too. Especially considering that the investigation really regards only one of those theories.

Response 21

We thoroughly re-structured the Discussion. As we only have the results in body size, so we accented this part and separated it from other hypotheses discussed in Discussion solely on the literature data.

Comment 22

LL378-379 Would the authors consider to rephrase 'Nevertheless, pup calls have clear species-specific differences in the acoustics' in 'Nevertheless, pup calls show clear species-specific differences in their acoustic features'?

Response 22

This sentence was corrected as recommended (Lines 396-397).

Comment 23

LL379-380 The relationship between this sentence and the previous one is unclear.

Response 23

To relate better this sentence and the previous text, we re-wrote it as follows (Lines 397-399): "The acoustic adaptation hypothesis was considered here in the first time in application to

vocalizations of pup rodents, whereas previously it was only applied for explaining the evolution of species-specific vocalizations in adult rodents [104-107].”

Comment 24

LL382 Missing comma after 'elaboration' but the whole sentence at LL381-383 is not very clear. Please, consider to rephrase it.

Response 24

We deleted “elaboration” to simplify this sentence.

Comment 25

LL384-385 'We can consider this hypothesis in application to our data, in spite of difficulties with estimating the complexity of the vocal repertoire in pup gerbils'. Can the authors add a reference?

LL386-387 We can estimate the complexity of the vocal repertoire via the complexity of the acoustics of USV calls (contour shape, percentage of vocal nonlinear phenomena). As in the previous comment, can the authors add a reference?

Response 25

This kind of study was done in the first time, so there are no references to support this point. So, we combined this sentence with the following one in one sentence (Lines 403-405): “In this study, we can estimate the complexity of the vocal repertoire of pup gerbils *via* the complexity of the acoustics of USVs (contour shape, percentage of vocal nonlinear phenomena).”

Comment 26

LL402-403 'In contradiction to the social complexity hypothesis, the most complex pup USV calls with most variable contour shapes were found in *Dipodillus campestris* (Fig 4)...' I am not really sure about referring to figures (or tables), within the discussion (see also L423, L426, L427...).

Response 26

This is commonly applied approach; we saw this in many papers.

Comment 27

LL404-405 'Consistently, USV calls complicating with nonlinear phenomena, were mostly frequent in the medium-social *Pachyuromys duprasi*.' The whole sentence is a bit unclear. 'Consistently' with the same hypothesis? Maybe 'complicated by' instead of 'complicating with'?

Response 27

Was corrected as recommended (Line 421).

Comment 28

L416 Not really sure about using 'quantitative differences'

Response 28

Was replaced with “Regarding the differences in the acoustic variables” (Line 433).

Comment 29

LL426 'Call duration, $f_{0\text{beg}}$ and f_{peak} differ in *Meriones unguiculatus*...'

Missing comma after $f_{0\text{beg}}$; moreover, I would use the extended names (i.e. initial f_0).

Response 29

We added the missing comma and used the extended names (Line 443): "Call duration, the f_0 at the onset of a call, and the peak frequency differ".

Comment 30

LL429-432 'The early study for call-based taxonomy of *Pachyuromys duprasi* and five species of genus *Meriones* [25] also reports that similarities and dissimilarities of vocalizations

(ultrasonic and audible) between these gerbil taxa are not sufficient for any reliable conclusions about the relationship between the acoustics and phylogeny'. Maybe 'An early study' instead of 'The early study'; 'among' instead of 'between'. However, would the author consider to rephrase the whole sentence?

Response 30

We re-phrased this sentence as follows (Lines 446-449): “An early study for call-based taxonomy of *Pachyuromys duprasi* and five species of genus *Meriones* [25] also reports that similarities and dissimilarities of vocalizations (ultrasonic and audible) among these gerbil taxa are not sufficient for elucidating the relationship between the acoustics and phylogeny.”

Comment 31

LL453-455 ‘The effect of species-specific characteristics on the acoustics of USV calls exceeded significantly the effect of body size’. What do the authors mean with 'species-specific characteristics'?

Response 31

We re-wrote this sentence as (Lines 378-379): “The effect of species on the acoustics of USVs exceeded significantly the effect of body size (Table 4).”

Comment 32

LL456-462 Do not really understand why the conclusion addresses the audible calls, all of a sudden. It can surely represent a further direction but I would prefer a conclusion about the USV calls rather than the audible ones.

Response 32

This part was deleted during re-structuring the Discussion.

Reviewer: 2

Comments to the Author(s)

You have presented a novel and interesting study on the species differences in infant Gerbil isolation calls. Extensive data collection and acoustical analyses have been undertaken to characterise these calls, so you should be commended. I have some queries and suggestions over the statistical analyses and interpretation of results. I would additionally suggest to have the English of this paper edited.

More specifically:

Comment 33

- In the abstract, please mention what vocal variables were measured. E.g. pitch, temporal parameters etc.

Response 33

We added in Abstract which acoustic variables were measured (Line 41): "(duration, fundamental and peak frequency)..."

Comment 34

-Line 49 - please clarify the statement starting with "the effect of species identity..."

Response 34

We re-wrote this sentence as follows (Lines 48-49): “The effect of species identity on the acoustics was stronger than the effect of body size.”

Comment 35

- If you report on average values within the manuscript, you need to specify this in your abstract, as it looks like you have extracted this information from the entire 1200 calls rather than collapsing it down.

Response 35

We added in Abstract: “the average values” (Line 40).

Comment 36

Introduction: All in all, this section could be written more concisely. A lot of basic background information was provided on the species, and further synthesis is needed. The gap in knowledge also needs to be more distinctly highlighted.

Response 36

We slightly re-wrote the Introduction to make it more concise. We also tried to better highlight the gap of knowledge (Lines 95-96): “For pup gerbils, USVs have yet to be examined in cross-species perspective and for their potential relationship between f_0 and body size.”

Comment 37

Please correct statement to "Gerbils or jirds are".

Response 37

Was corrected (Line 56).

Materials and methods:

Comment 38

- Some of the information provided on the gerbil test subjects may be better presented in tabular form. E.g. species, number of subjects, origin, year of data collection, age of subjects.

Response 38

We shortened the information about the origin of the colonies (Lines 122-130, please see Response 6). Other information can be presented more concisely in the textual form, as it is common for many species and should not be repeated in the tabular form.

Comment 39

- How did you determine which pups to select in each litter? Was it randomized or balanced for sex?

Response 39

We added in text (Line 142): “Pups were unsexed, in each litter pups were selected randomly.”

Comment 40

- Line 164 – with minimal reverberation.

Response 40

Was corrected as recommended (Lines 161-162, see Response 7).

Comment 41

- Line 177 – to minimise the manipulations on the pup during the trial.

Response 41

Was corrected as recommended (Line 174).

Comment 42

- Line 191 does not make sense. Successively would be a better word than consequently.

Response 42

"consequently" was replaced with "one by one" (Line 188)

Comment 43

- Line 199 – 204 – sentence a bit hard to follow. Please rephrase.

Response 43

We re-phrased this sentence (Lines 196-201, please see Response 8).

Comment 44

- Do you know how many calls in total were there to select from?

Response 44

It could be from few dozens to many hundreds calls per file.

Comment 45

- Line 226 – according to.

Response 45

Was corrected as recommended (Line 224)

Comment 46

- Please confirm - Was it a single observer conducting the classification of the f0 contour shapes?

Response 46

We added in the text (Lines 225-226): “One researcher (JDK) classified the calls and another researcher (IAV) confirmed this classification.”

Comment 47

Statistical analyses – there are updated techniques for incorporating repeated measures into your statistical analyses. Given that your dataset is of adequate size, I would recommend using one of these approaches rather than taking the average for individual subjects. Instead of Anova you could try a linear mixed effect model, accounting for individual pup as a random effect.

Response 47

The aim of this study is the comparison of the acoustics of pup isolation calls among six species. We deliberately removed the factor individuality from the analysis, by using the average values of acoustic variables per individual. Comparison of vocal individuality has yet to be done as a separate study.

We conducted the recommended analyses: linear mixed effect model, individual pup nested in species, with species - fixed factor and individual - random factor. The results of comparison among species differ from those presented in Table 3 (using one-way ANOVA) only towards a small increase of differences, what is the effect of increase of the number of degrees of freedom because of the multiple measurements from the same individual (pseudoreplication).

Comment 48

- Further, seeing as you have collected 1200 calls, it would be interesting to see the DFA classification percentages when the entire dataset is used. See Favaro et al., 2016. Favaro, L.; Gili, C.; Da Rugna, C.; Gnone, G.; Fissore, C.; Sanchez, D.; McElligott, A. G.; Gamba, M.; Pessani, D. Vocal Individuality and Species Divergence in the Contact Calls of Banded Penguins. *Behav. Processes* 2016, 128, 83–88. <https://doi.org/10.1016/j.beproc.2016.04.010>. I think this approach would be more appropriate and would reduce your classification accuracy (which is extremely high).

Response 48

In this study, we focused exactly on the differences among the species and do not want to introduce the effect of pseudoreplication in DFA. Yes, all 100% of correct assignment (for 6 groups and 60 samples) is surprising and occurs in the first time in my practice. But this result can be re-checked based on the raw data from Table 1S

Comment 49

- For your current GLMMs, what was the family that you used? Further, please specify your random effects.

Response 49

We used the Gaussian family, because "The values were normally distributed for all body size and acoustic variables (Kolmogorov–Smirnov test)" (Lines 249-250).

Comment 50

- Please mention the number of acoustic variables measured in the stat analysis.

Response 50

We indicated the number of acoustic variables measured in the stat analysis in Statistical analyses section (Lines 247-248 and 262).

Comment 51

- For the DFA, was a cross- validation procedure undertaken?

Response 51

We performed a cross-validation leave-one-out, as was recommended. We added in the Methods (Lines 263-264): "We performed a cross-validated (leave-one-out) DFA to determine if USVs could be correctly classified to the correct species." We added in the Results (Lines 353-355): "The accuracy of the DFA decreased to 98.3% when the more conservative leave-one-out cross-validation was applied. Only one single USV of *Meriones unguiculatus* was incorrectly assigned to *Gerbillus perpallidus*."

Comment 52

- I would suggest conducting a Manova as well.

Response 52

Please see Response 47

Results:

Comment 53

- Line 298 - present not usual.

Response 53

This correction changes the sense, so we kept as it is.

Comment 54

- Body size PCA -I would also highlight the high eigenvalue of the first PC compared to the other two.

Response 54

Eigenvalues for all the three PCA factors are presented in Table 2.

Comment 55

- DFA - mention which variables are loaded onto which discriminant functions.

Response 55

We added in MS the Table 5 with values of correlation between acoustic variable of ultrasonic calls and the five discriminant functions, eigenvalues and percent variances, described by each function. We added in the text (Lines 357-362): "The first discriminant function described 75.45% of the variance and was correlated with all fundamental frequency parameters (f0beg, f0max, f0end, f0min) and peak frequency (Table 5). The second discriminant function described 12.73% of the variance and correlated only with f0end. The third discriminant function described 8.93% of the variance and strongly correlated with duration. The fourth and fifth discriminant functions described only 2.89% of the variance (Table 5)."

Comment 56

- Figure 5 – it is difficult to see the percentages of biphonation and frequency jumps for some species. Could you instead graph the proportion of biphonation and fj in the species where NLP is present?

Response 56

All percentages of biphonation and frequency jumps for each species are given in the text (Lines 301-308).

Comment 57

- Table S2 - continuous factors not continual.

Response 57

Corrected to "continuous"

Comment 58

- Have you considered classifying the calls to the correct individual within each species? i.e. conducting 6 separate DFAs. This would help confirm or refute your species classification percentages.

Response 58

The aim of this MS is to compare the acoustics of pup isolation calls among six species. We deliberately removed the factor individuality from the analysis. Comparison of the individualistic traits in calls is planned to be done in frames of another study.

Discussion:

Comment 59

- You mention the presence of NLP and the qualitative distinctiveness of *pachyuromys duprasi*. But I cannot see any inferential stats for the NLP and USV contour shapes. Could you not do a binomial GLMM to compare the presence or absence of NLP between species? Further, a multinomial model to compare the usv shapes amongst the species? This would further support your claims.

Response 59

We added in the Methods (Lines 258-259): "We used binomial GLM to compare the presence or absence of nonlinear vocal phenomena between species." We added in the Results (Lines 299-301): "Binomial GLM showed that species identity significantly affected the presence or absence of nonlinear vocal phenomena in pup USVs (estimate=0.472±0.076, $z=6.24$, $p<0.001$)."

Comment 60

- It is a bit hard to keep track of all the hypotheses, especially when your results do not align with some of them. I'd suggest focusing on the more relevant and relatable hypotheses.

Response 60

We think that it is important to keep this discussion with these commonly considered hypotheses. Before considering them, we cannot know whether they are more or less relevant to our data.

Appendix C

Reviewer comments to Author:

Reviewer: 2

Thank you for reviewing my manuscript suggestions and providing detailed answers to them. I still have some minor queries about the statistical approach, which once answered, I will be happy to accept the manuscript. Please see below:

Comment 1

Line 124: Some typos - please amend.

Response 1

Was corrected

Comment 2

Line 251: You previously mentioned that you log transformed the body weight variable. This should also be mentioned here - that you log transformed to satisfy this assumption of normality.

Response 2

We log transformed the body weight as proxy of linear body size, not to satisfy the assumption of normality. We re-wrote the sentence to make it clear (Lines 185-187): "The body variables and log body weight were taken as proxies of body size for further comparison with the USV acoustic variables."

Comment 3

Line 254: Tukey's Honest Significant Difference.

Response 3

Was corrected

Comment 4

Line 254: There is plenty of literature which argues that the mixed method approach is more appropriate for dealing with repeated measures and avoiding type 1 errors associated with pseudoreplication. Please clarify/correct paragraph starting on line 254 as this part is still unclear. Please justify choice of one-way ANOVA with averaged values, as opposed to a mixed-effect model with the full dataset, accounting for repeated measures using individual ID as a random effect. You mention in your response that you conducted the LMM, so these should be reported, along with the estimated marginal means for each species.

Response 4

We did not include the analyses of vocal individual identity in the MS, because the aim of this study was in the comparison of pup ultrasonic isolation calls among six gerbil species. We conducted the linear mixed effect model (individual pup nested in species, with species - fixed factor and individual - random factor) only to be convinced that the results of comparison among species do not differ from those presented in Table 3. However, the statistical analysis used in MS (comparison of averaged values of acoustic variables per individual) is more appropriate for the purpose this study (comparison among species, not among individuals within species). The applied analysis avoids multiple measurements of acoustic variables from the same individual (pseudoreplication). Moreover, it is more correct as one averaged acoustic measurement per individual corresponds to one measurement of each body variable per individual (see Table 3). Furthermore, decrease of the number of degrees of freedom makes the results more robust. We re-wrote the text (Lines 249-254): "For each individual subject, the averaged values of six acoustic variables over 20 calls were used for the statistical comparisons. This allowed to avoid multiple measurements of acoustic variables from the same individual (pseudoreplication), to match one averaged acoustic measurement per individual with one measurement of each body variable per individual and to decrease the number of degrees of freedom for more robust results."

Comment 5

Line 258/comment 49: You have mentioned that you used a GLMM, but you have not mentioned what the random effect in this model is. Please clarify. Did you include individual identity as a random effect? Were species and body size included in the same model, with acoustic parameter as the outcome? Accordingly, I am still confused by some of the results, e.g. Table 4. Why not conduct post-hoc comparisons for the GLMM instead of the ANOVA in table 3?

Response 5

We used GLMM, as pup species was included in analysis as a categorical factor and index body size was included as a continuous factor. This is a mixed model, as the factors are unequal. We did not include individual identity as a random effect (please see a previous response).

We included species and body size in the same model, with acoustic parameter as the outcome.

We indicated in the Methods (Lines 259-262): “We used Generalized Linear Mixed Model (GLMM) with Tukey HSD test for estimating the effects of pup species and pup body size on the acoustic variables of USVs, with pup species as a categorical factor and index body size as a continuous factor.”

We deleted the results of one-way ANOVA and concentrated all results for species identity and body size index effects on the acoustic variables of ultrasonic calls of six Gerbillinae species in Table 3. Table 4 was deleted. Results of post hoc Tukey HSD test we the same for one-way ANOVA and for GLMM.

Comment 6

Line 260: Specify response and categorical factors for the binomial GLM.

Response 6

We wrote (Line 262-265): “We used binomial GLM to compare the presence or absence of nonlinear vocal phenomena between species, with pup species as a categorical factor and presence or absence of nonlinear vocal phenomena as a response factor.”

Comment 7

Line 301/comment 53: Usual still does not really make sense here. Common a better phrase.

Response 7

Was corrected as recommended

Comment 8

Line 348: Correlated significantly negatively a bit awkward. Rephrase.

Response 8

Was corrected to “significantly negatively correlated”

Comment 9

Line 356: Typo. Single

Response 9

Was corrected

Comment 10

Table 5: Rename root as discriminant function 1, df2... etc.

Response 10

Was corrected